# Early Stage Convergence and Global Convergence of Training Mildly Parameterized Neural Networks

**Mingze Wang**
School of Mathematical Sciences
Peking University
Beijing, 100081, P.R. China
mingzewang@stu.pku.edu.cn

**Chao Ma**
Department of Mathematics
Stanford University
Stanford, CA 94305
chaoma@stanford.edu

## Abstract

The convergence of GD and SGD when training mildly parameterized neural networks starting from random initialization is studied. For a broad range of models and loss functions, including the most commonly used square loss and cross entropy loss, we prove an "early stage convergence" result. We show that the loss is decreased by a significant amount in the early stage of the training, and this decrease is fast. Furthurmore, for exponential type loss functions, and under some assumptions on the training data, we show global convergence of GD. Instead of relying on extreme over-parameterization, our study is based on a microscopic analysis of the activation patterns for the neurons, which helps us derive more powerful lower bounds for the gradient. The results on activation patterns, which we call "neuron partition", help build intuitions for understanding the behavior of neural networks' training dynamics, and may be of independent interest.

## 1 Introduction

As deep learning shows its capability in various fields of applications, extensive researches are done to theoretically understand and explain its great success. Among the topics covered by these theoretical studies, the optimization of deep neural networks is one of the most crucial one, especially given the fact that simple optimization algorithms such as Gradient Descent (GD) and Stochastic Gradient Descent (SGD) can easily achieve zero training loss (Zhang et al., 2017), although the loss landscape of training neural networks is highly non-convex (Li et al., 2018; Sun et al., 2020). Existing works attempting to answer the surprising convergence ability usually work in settings that do not align well with realistic practices. For example, the notable Neural Tangent Kernel (NTK) theory (Jacot et al., 2018; Du et al., 2018; 2019; Allen-Zhu et al., 2019b; Zou et al., 2018) shows the global convergence of SGD in highly over-parameterized regime, in which the loss landscape is approximated by a quadratic function. In practice, though, the loss landscape is indeed highly non-convex with spurious local minima and saddle points (Safran and Shamir, 2018; Safran et al., 2021), and the dynamics of neurons show nonlinear behaviors (Ma et al., 2020).

Nevertheless, fast decreasing of the loss value always happens when the network trained is not highly over-parameterized, at least in the early stage of training. It is common that the loss experiences a drastic decreasing at the beginning of the training. In many cases, this decrease of loss even continues until the loss achieves zero, i.e. the optimization algorithm fully converges. In this paper, we study the early stage or full convergence of GD and SGD when under-parameterized or mildly over-parameterized models are trained. Specifically, we answer the following two theoretical questions:

*When we train practical-size neural networks by GD or SGD,*

*1. Does the fast convergence in the early stage of the training provably exist? If so, how*

36th Conference on Neural Information Processing Systems (NeurIPS 2022).

*long will the phenomenon last and how much will loss descend in the early stage?*
*2. Can the global convergence be proved under some special conditions*
*on loss function and training data?*

Our answers to the questions above are roughly summarized in the following main theorems:

**Main Theorem 1** (Informal statement of Theorem 4.2 and 4.6)**.** *For mildly over-parameterized or under-parameterized two-layer neural networks with quadratic loss or general classification loss (Assumption 4.4), let parameters be trained by GD or SGD started with random initialization. If the learning rate $\eta \leq 0.01$, then in the first $\Theta(1/\eta)$ iterations, the loss will descend $\Omega(1)$.*

**Main Theorem 2** (Informal statement of Theorem 5.2 and 5.3)**.** *For mildly over-parameterized or under-parameterized two-layer neural networks with exponential-type loss (Assumption 5.1), let parameters be trained by GD with random initialization and proper learning rate. If data is well-separated (satisfies Assumption 4.1), the loss converges to $0$ at exponential rate or arbitrary polynomial rate, depending on the conditions.*

In the first main theorem, we demonstrate that the fast convergence in the early stage of training happens under weak conditions. The model does not need to be highly over-parameterized, and the loss function can take quite general forms. In the second main theorem, we further prove the full convergence of GD for exponential-type loss and well-separated dataset. These assumptions on the loss function and training data are close to practice and widely picked by previous theoretical studies (Ji and Telgarsky, 2018; 2020; Phuong and Lampert, 2020; Chatterji et al., 2021b;a; Lyu and Li, 2019; Lyu et al., 2021). "Full convergence" means that our analysis covers all the stages of the training process, starting from initialization to the convergence to zero loss. This is different from existing works which focus on the convergence process after the training accuracy hits 100% (Lyu and Li, 2019; Ji and Telgarsky, 2020; Chatterji et al., 2021b). Moreover, we provide the convergence rate of GD for two classes of neural networks.

During the theoretical analysis, we study the dynamics of neurons in detail and capture the effect of each sample on each neuron. We call our results "neuron partition", which provides a accurate description to the behavior of neurons. With the neuron partition results, we derive a novel gradient lower bound for mildly parameterized neural networks, which is vital for the convergence results. The neuron partition also provides rich intuition for how the convergence happens. When GD or SGD is used with small initialization, the network is initially close to the saddle point at $0$, which might be hard for GD to escape. However, neurons will adjust their directions rapidly and enter a good region which contains neither spurious local minima nor saddle points. Then, neurons will keep moving through the right directions for $\Theta(1/\eta)$ iterations, during which the loss descends quickly and significantly. For exponential-type loss and well-separated training data, the second stage will proceed uninterruptedly until training is stopped.

## 2   Related Work

Due to the highly non-convex loss landscape (Li et al., 2018; Sun et al., 2020), classical optimization theories such as convex optimization fail to characterize the convergence of training neural networks. Researchers have proposed theories specifically for neural networks. We list some below.

A popular line of works focus on the highly over-parameterized neural networks and derive the "Neural tangent kernel (NTK)" theory. In the NTK regime (Jacot et al., 2018; Du et al., 2018; 2019; Allen-Zhu et al., 2019a;b; Arora et al., 2019a;b; Daniely, 2017; Zou et al., 2018; Li and Liang, 2018; Chizat et al., 2019; E et al., 2019; 2020), the neural network model is close to a kernel method, leading to the nearly convex optimization landscape. Global convergence is proven in this regime. Besides, for highly over-parameterized neural networks, the mean-field approach (Chizat and Bach, 2018; Mei et al., 2018; 2019) is another line, which analyze the training dynamics by Wasserstein gradient flow.

However, the NTK regime is different from real practical neural networks in several aspects. First, practical neural networks are often mildly over-parameterized rather than highly over-parameterized (Livni et al., 2014). Second, the landscape for practical neural networks is usually more complicated, containing local minima and saddle points (Safran and Shamir, 2018; Safran et al., 2021; Ding et al., 2019). Finally, the empirical superiority of neural networks over kernel methods is obvious. For

example, neural networks can learn single neuron efficiently (Yehudai and Ohad, 2020; Wu, 2022), while kernel methods fail unless the network size is exponentially large with respect to the input dimension (Yehudai and Shamir, 2019).

Furthermore, optimization theories for neural networks beyond the NTK regime has also been studied. Safran and Shamir (2018); Ding et al. (2019) proved that spurious local minima are common in the loss landscape. Safran et al. (2021) pointed out that neither one-point convexity nor Polyak-Łojasiewicz condition holds near the global minimum. In (Lyu and Li, 2019; Ji and Telgarsky, 2020; Chatterji et al., 2021b;a; Zhou et al., 2021), local convergence results are given with cross-entropy loss or some special networks at the late stage of training, i.e. when training loss is small enough. In (Li et al., 2020; Phuong and Lampert, 2020; Lyu et al., 2021; Jentzen and Riekert, 2021; Cheridito et al., 2022), global convergence results are given for some special networks, special data distribution or special target functions.

## 3 Preliminaries

### 3.1 Notations

We use bold letters for vectors or matrices and lowercase letters for scalars, e.g. $\boldsymbol{x} = (x_1, \cdots, x_d)^\top \in \mathbb{R}^d$ and $\mathbf{P} = (P_{ij})_{m_1 \times m_2} \in \mathbb{R}^{m_1 \times m_2}$. We use $\langle \cdot, \cdot \rangle$ for the standard Euclidean inner product between two vectors, and $\|\cdot\|$ for the $l_2$ norm of a vector or the spectral norm of a matrix. We use $a \lesssim b$ to indicate that there exists an absolute constant $c > 0$ such that $a \leq cb$, and $a \gtrsim b$ is similarly defined. We use standard progressive representation $\mathcal{O}, \Omega, \Theta$ to hide absolute constants. For any positive integer $n$, let $[n] = \{1, \cdots, n\}$. Denote by $\mathcal{N}(\boldsymbol{\mu}, \boldsymbol{\Sigma})$ the Gaussian distribution with mean $\boldsymbol{\mu}$ and covariance matrix $\boldsymbol{\Sigma}$, $\mathbb{U}(S)$ the uniform distribution on a set $S$. Denote by $\mathbb{I}\{E\}$ the indicator function for an event $E$. For a square matrix $P$, we use $\lambda_{\min}(\mathbf{P})$ to denote its smallest singular value.

### 3.2 Problem settings

In this paper, we consider supervised learning problems. Let $\mathcal{P}$ be a data distribution on $\mathbb{R}^d \times \mathbb{R}^C$. We are given $n$ training data $\mathcal{S} = \{(\boldsymbol{x}_i, \boldsymbol{y}_i)\}_{i=1}^n \subset \mathbb{R}^d \times \mathbb{R}^C$ drawn i.i.d. from $\mathcal{P}$. Without loss of generality, we assume $\|\boldsymbol{x}\| \leq 1$ for any $(\boldsymbol{x}, \boldsymbol{y})$ sampled from $\mathcal{P}$.

We consider the empirical risk minimization (ERM) problem, which tries to minimize the empirical risk $\mathcal{L}(\cdot)$ with loss function $\ell(\cdot, \cdot)$:

$$\mathcal{L}(\boldsymbol{\theta}) = \frac{1}{n} \sum_{i=1}^n \ell(\boldsymbol{y}_i, \boldsymbol{f}(\boldsymbol{x}_i; \boldsymbol{\theta})), \tag{1}$$

where $\boldsymbol{f}(\boldsymbol{x}; \boldsymbol{\theta}) \in \mathbb{R}^C$ is the model and $\boldsymbol{\theta}$ represents all parameters of the model. We will give specific forms to the loss function and the model in the analysis in later sections.

We use Gradient Descent (GD) or Stochastic Gradient Descent (SGD) starting from random initialization to solve the ERM problem. The update rules of GD or batch SGD can be written as:

$$\textbf{GD}: \quad \boldsymbol{\theta}(t+1) = \boldsymbol{\theta}(t) - \eta_t \nabla \mathcal{L}(\boldsymbol{\theta}(t)), \tag{2}$$

$$\textbf{SGD}: \quad \boldsymbol{\theta}(t+1) = \boldsymbol{\theta}(t) - \frac{\eta_t}{B} \sum_{i \in \mathcal{B}_t} \nabla \ell(\boldsymbol{y}_i, f(\boldsymbol{x}_i; \boldsymbol{\theta}(t))), \tag{3}$$

where $\mathcal{B}_t = \{\gamma_{t,1}, \cdots, \gamma_{t,B}\}$ is a batch, and $\gamma_{t,1}, \cdots, \gamma_{t,B} \overset{\text{i.i.d.}}{\sim} \mathbb{U}([n])$ and are independent with $\boldsymbol{\theta}(t)$. The random initialization will be specified in each theorem.

## 4 Early Stage Convergence

In this section, we state and discuss our main results on the fast convergence in the early stage of training. The models we focus on are mildly over-parameterized or under-parameterized neural networks. As a warm up, in Section 4.1, we study a simple case with binary classification problems and quadratic loss. Then, in Section 4.2, we extend our results to multi-class classification problems with general loss and one-hot labels. We provide discussions to our results in Section 4.3.

## 4.1 Binary classification with quadratic loss

In this subsection, we study the ERM problem (1) with quadratic loss $\ell(y_1, y_2) = \frac{1}{2}(y_1 - y_2)^2$ and the two-layer ReLU neural network model without bias:

$$f(\boldsymbol{x}; \boldsymbol{\theta}) = \sum_{k=1}^{m} a_k \sigma(\boldsymbol{b}_k^\top \boldsymbol{x}), \tag{4}$$

where $\sigma(z) = \mathrm{ReLU}(z)$, $\boldsymbol{\theta} = (a_1, \cdots, a_m, \boldsymbol{b}_1^\top, \cdots, \boldsymbol{b}_m^\top)^\top \in \mathbb{R}^{dm+m}$. We consider the following random initialization

$$a_k(0) \overset{\text{i.i.d.}}{\sim} \mathbb{U}(\pm 1/\sqrt{m}), \quad \boldsymbol{b}_k(0) \overset{\text{i.i.d.}}{\sim} \mathcal{N}(\boldsymbol{0}, \frac{\kappa^2}{md}\mathbf{I}_d), \quad \text{for } k \in [n], \tag{5}$$

where $\kappa$ is a constant that controls the initial scale of the first layer.

We focus on the training data with two classes with good separation given by the following assumption.

**Assumption 4.1.** (i) $n$ is even. $y_i = 1$ for $i \in [n/2]$; $y_i = -1$ for $i \in [n] - [n/2]$. $\boldsymbol{x}_i^\top \boldsymbol{x}_j \geq 0$ for $i, j$ in the same class; $\boldsymbol{x}_i^\top \boldsymbol{x}_j \leq 0$ for $i, j$ in different classes. (ii) There exists a constant $\mu_0 > 0$, s.t.

$$\min_{\{i \in [n], \boldsymbol{v} \in \mathbb{R}^d | \boldsymbol{v}^\top \boldsymbol{x}_i \leq 0\}} \max_{\{j \in [n] | \boldsymbol{v}^\top \boldsymbol{x}_j > 0\}} y_i \boldsymbol{x}_i^\top \boldsymbol{x}_j y_j \geq \mu_0.$$

For Assumption 4.1 (i), similar assumption has been used in prior works (Phuong and Lampert, 2020). For image classification problems, this assumption can be ensured by a simple transformation on the data due to the non-negative pixels of images. Specifically, given an image dataset consisted of two classes, we just need to replace $\boldsymbol{x}$ with $-\boldsymbol{x}$ for any $(\boldsymbol{x}, y)$ in the second class. Assumption 4.1 (ii) is a technical assumption used in the analysis. It is not a strong addition over Assumption 4.1 (i). For example, if there exists a $(x_0, y_0)$ in the dataset $S$ such that $(-x_0, -y_0)$ is also in $S$, then the dataset $S$ satisfies Assumption 4.1 (ii) with $\mu_0 = 1$ (the upper bound of $\mu_0$ is 1). Admittedly, this technical assumptions is a limitation of our theory—it restricts its applicability. We will search for the relaxation of this assumption in future works.

Now we state the following result.

**Theorem 4.2.** *Suppose Assumption 4.1 holds. Let $\boldsymbol{\theta}(t)$ be the parameters of model (4) trained by Gradient Descent (2) with quadratic loss. If the width $m = \Omega(\log(n/\delta))$, the input dimension $d = \Omega(\log m)$, the learning rate $\eta_t = \eta \leq 0.01$ and the initialization scale $\kappa = \mathcal{O}(\eta\mu_0/n)$ in (5), then, with probability at least $1 - \delta - 2me^{-2d}$, the loss will descend $\Omega(1)$ in $T = \Theta(\frac{1}{\eta})$ iterations.*

From our proof in Appendix A, we have the following corollary with fixed numbers about $\eta, \kappa, m$ instead of progressive expressions.

**Corollary 4.3.** *Suppose Assumption 4.1 holds. Let $\boldsymbol{\theta}(t)$ be the parameters of model (4) trained by Gradient Descent (2) with quadratic loss. Let $\eta = \mathbf{0.01}$, $\kappa \leq \min\{\eta/2000, \eta\mu_0/3n\}$, $m \geq \max\{144\log(2n^2/\delta), 4\}$, and $\gamma_1, \gamma_2 = \Theta(1)$ be constants related with the data distribution defined in (10) in Appendix A. Then, with probability at least $1 - \delta - 2me^{-2d}$, in $T = \mathbf{44}$ iterations, loss will descend at least $0.193(\gamma_1 - \gamma_2\sqrt{8\log(n^2/\delta)/m}) - 0.0111 = \Omega(1 - \sqrt{\log(n/\delta)/m})$.*

Conclusions in Theorem 4.2 show the meaning of **"mildly parameterized"** in our title. The network does not need to be over-parameterized, i.e. the number of parameters may be smaller than the number of training data. Hence, we pick the term "mildly parameterized" instead of the more widely use "mildly over-parameterized". The former is weaker than the latter.

## 4.2 Multi-class classification with general loss and one-hot labels

In this subsection, we extend the early stage convergence results above to more practical settings. Specifically, we consider classification problem with one-hot labels and the general loss functions that satisfy the following conditions.

**Assumption 4.4.** The loss function $\ell(\cdot, \cdot)$ can be expressed as $\ell(\boldsymbol{y}_1, \boldsymbol{y}_2) = \tilde{\ell}(\boldsymbol{y}_1^\top \boldsymbol{y}_2)$ such that: (i) $\tilde{\ell}(\cdot)$ is twice differentiable in $\mathbb{R}$; (ii) $\tilde{\ell}(\cdot)$ is non-negative and non-increasing in $\mathbb{R}$; (iii) there exist $z_0 \in (0, 1]$, $g_{\min}, g_{\max} > 0, h_{\max} \geq 0$ such that $g_{\min} \leq -\tilde{\ell}'(z) \leq g_{\max}$ and $0 \leq \tilde{\ell}''(z) \leq h_{\max}$ hold for any $z \in [0, z_0]$.

It is easy to verify that exponential loss $\ell(\boldsymbol{y}_1, \boldsymbol{y}_2) = e^{-\boldsymbol{y}_1^\top \boldsymbol{y}_2}$ ($z_0 = 1$, $g_{\min} = \frac{1}{e}$, $g_{\max} = 1$, $h_{\max} = 1$), logistic loss $\ell(\boldsymbol{y}_1, \boldsymbol{y}_2) = \log(1 + e^{-\boldsymbol{y}_1^\top \boldsymbol{y}_2})$ ($z_0 = 1$, $g_{\min} = \frac{1}{e+1}$, $g_{\max} = \frac{1}{2}$, $h_{\max} = \frac{1}{4}$) and hinge loss $\ell(\boldsymbol{y}_1, \boldsymbol{y}_2) = \max\{0, 1 - \boldsymbol{y}_1^\top \boldsymbol{y}_2\}$ ($z_0 = 1$, $g_{\min} = 1$, $g_{\max} = 1$, $h_{\max} = 0$) all satisfy the conditions in Assumption 4.4.

On the training data, we impose the following assumption, which is milder than Assumption 4.1.

**Assumption 4.5.** There exists $s > -1$ s.t. $\langle \boldsymbol{x}_i, \boldsymbol{x}_j \rangle \geq s$ for any $i, j \in [n]$.

Note that now we do not need any separability of training data, but only require there is no pair of data going into opposite directions. (We can even normalize the data and let $\|\boldsymbol{x}\| < 1 - \epsilon$ for some $\epsilon > 0$, rather than $\|\boldsymbol{x}\| \leq 1$. Then, the above assumption naturally holds.) Notably, Assumption 4.5 is more widely applicable than Assumption 4.1 above. It holds for normalized image datasets such as MNIST (LeCun et al., 1998) and CIFAR-10 (Krizhevsky et al., 2009).

For the model, we use the two-layer ReLU neural network model with bias as our prediction model:

$$\boldsymbol{f}(\boldsymbol{x}; \boldsymbol{\theta}) = \sum_{k=1}^{m} \boldsymbol{a}_k \sigma(\boldsymbol{b}_k^\top \boldsymbol{x} + c_k), \tag{6}$$

where $\sigma(z) = \text{ReLU}(z)$, $\boldsymbol{a}_k \in \mathbb{R}^C$ and $\boldsymbol{\theta} = (\boldsymbol{a}_1^\top, \cdots, \boldsymbol{a}_m^\top, \boldsymbol{b}_1^\top, \cdots, \boldsymbol{b}_m^\top, c_1, \cdots, c_m)^\top \in \mathbb{R}^{(d+C+1)m}$. We consider random initialization $\boldsymbol{b}_k(0) \overset{\text{i.i.d.}}{\sim} \mathcal{N}(\boldsymbol{0}, \frac{\kappa^2}{m(d+1)} \mathbf{I}_d)$, $c_k(0) = \frac{\kappa}{\sqrt{m(d+1)}}$, and $\boldsymbol{a}_k(0) = (1/\sqrt{m}, \cdots, 1/\sqrt{m})^\top$ for $k \in [m]$, where $\kappa$ is again a constant that controls the initial scale of the first layer.

We show the following convergence result:

**Theorem 4.6.** *Under Assumption 4.4 and 4.5, let $\boldsymbol{\theta}(t)$ be the parameters of model (6) trained by Stochastic Gradient Descent (3). If the width $m = \Omega(1)$, the input dimension $d = \Omega(\log m)$, the batch size $B = \Omega(\log m)$, the learning rate $\eta_t = \eta \leq 0.01$ and the initialization scale $\kappa = \mathcal{O}(\eta/B)$, then with probability at least $1 - \delta - \mathcal{O}(me^{-d}) - \mathcal{O}(m0.17^B)$, the loss will descend $\Omega(1)$ in $T = \Theta(\frac{1}{\eta})$ iterations.*

By our analysis in Appendix B, we have the following corollary which gives specific numbers about $\eta, \kappa, m$ instead of progressive expression.

**Corollary 4.7.** *Under the same assumptions as Theorem 4.6, let $\boldsymbol{\theta}(t)$ be the parameters of the model (6) trained by SGD (3). Let $\eta = \mathbf{0.01}$, $\kappa \leq \min\{\eta/10, \eta/3B\}$, $m \geq 6$; $z_0 = 1$, $g_{\min} = \frac{1}{2}$, $g_{\max} = 1$, $h_{\max} = 1$ (in Assumption 4.4) and $s = 0$ (in Assumption 4.5). Then, with probability at least $1 - \delta - 4me^{-\frac{d+1}{2}} - m0.17^B$, in $T = \mathbf{34}$ iterations, loss will descend at least $0.262533 = \Omega(1)$.*

We remark here that the weaker Assumption 4.5 in this subsection compared with the Assumption 4.1 before is made possible by the one-hot labels. For one-hot labels, each label component is non-negative. Hence, neural networks can quickly learn a positive bias, allowing the loss to drop significantly.

## 4.3 Discussion

A large number of classical convergence analysis focus on the late stage of training. For neural networks' optimization, many works focus on the period of training after the loss is smaller enough (Lyu and Li, 2019; Ji and Telgarsky, 2020; Chatterji et al., 2021b; Zhou et al., 2021). However, the early stage is at least as important as the late stage, especially for non-convex problems.

The theoretical results we present in this section demonstrate the fast convergence during the early stage of optimization, i.e. the (short) period of time after the initialization. The two main results, Theorem 4.2 and 4.6, imply that if we train two-layer neural networks by GD or SGD with learning rate $\eta \leq 0.01$, then the loss will descend significantly ($\Omega(1)$) in the first $T = \Theta(1/\eta)$ iterations.

Our results work under realistic settings. For models, our results hold for mildly over-parameterized and under-parameterized neural networks, because we only need the width $m = \Omega(\log(n/\delta))$. For loss functions, our results apply for practical losses, such as quadratic loss and cross entropy loss. On the algorithm side, the results work for both GD (2) and SGD (3).

Though both concerning convergence of GD and SGD, our results are essentially different from the NTK theory, especially on the requirement of over-parameterization. Our results only need $m = \Omega(\log(n/\delta))$, while in the NTK analysis usually assumes that $m$ is polynomial in $n, 1/\kappa$ or etc, excluding the practical settings. Moreover, from the analysis side, we establish fine-grained analysis of the training dynamics of each neuron, which may be of independent interested. See Section 6 for a summary of our techniques, and Appendix A and B for the detailed proof. Our analysis helps understand how the convergence in the early stage happens. When GD or SGD is used with small initialization, the initial network is close to the saddle point at 0. However, neurons will adjust directions rapidly, and the iterator will enter a good region which contains neither spurious local minima nor saddle points. Then, neurons will keep going towards the right directions for a period of time, during which process the loss will descend fast and significantly.

## 5    Global Convergence

When stronger conditions are imposed on the data distribution and the loss function, we can show global convergence of GD, i.e. starting from any initialization, the GD descends the loss to 0.

For loss functions, we consider the following exponential-type classification loss:

**Assumption 5.1.** The loss function $\ell(\cdot, \cdot)$ can be expressed as $\ell(\boldsymbol{y}_1, \boldsymbol{y}_2) = \tilde{\ell}(\boldsymbol{y}_1^\top \boldsymbol{y}_2)$ such that: (i) $\tilde{\ell}(\cdot)$ is twice continuously differentiable in $\mathbb{R}$; (ii) $\tilde{\ell}(\cdot)$ is positive and non-increasing in $\mathbb{R}$; (iii) there exist $g_b > 0, h \geq 0$ s.t. $-\frac{\tilde{\ell}'(z)}{\tilde{\ell}(z)} \leq g_b$ and $0 \leq \frac{\tilde{\ell}''(z)}{\tilde{\ell}(z)} \leq h$ for any $z \in \mathbb{R}$; (iv) there exists $g_a > 0$, s.t. $g_a \leq -\frac{\tilde{\ell}'(z)}{\tilde{\ell}(z)}$ for any $z \geq 0$.

The similar assumption on exponential-type loss has been used in prior works (Lyu and Li, 2019). It is easy to verify that both exponential loss $\ell(\boldsymbol{y}_1, \boldsymbol{y}_2) = e^{-\boldsymbol{y}_1^\top \boldsymbol{y}_2}$ $(g_a = g_b = h = 1)$ and logistic loss $\ell(\boldsymbol{y}_1, \boldsymbol{y}_2) = \log(1 + e^{-\boldsymbol{y}_1^\top \boldsymbol{y}_2})$ $(g_a = \frac{1}{2}, g_b = 1, h = 1)$ satisfy the conditions in Assumption 5.1.

Let $V$ be a constant defined as

$$V = \frac{1}{16}\Big(\frac{1}{2} - \sqrt{\frac{8\log(n^2/\delta)}{m}}\Big) \max\Big\{\frac{2}{n} + \frac{n-2}{n}\gamma, \lambda_{\min}(\mathbf{X}_+^\top \mathbf{X}_+) \wedge \lambda_{\min}(\mathbf{X}_-^\top \mathbf{X}_-)\Big\}, \quad (7)$$

where $\mathbf{X}_+ := (\boldsymbol{x}_1, \cdots, \boldsymbol{x}_{\frac{n}{2}}) \in \mathbb{R}^{d \times \frac{n}{2}}$, $\mathbf{X}_- := (\boldsymbol{x}_{\frac{n}{2}+1}, \cdots, \boldsymbol{x}_n) \in \mathbb{R}^{d \times \frac{n}{2}}$, and $\gamma := \min\limits_{i,j \text{ in the same class}} \boldsymbol{x}_i^\top \boldsymbol{x}_j \geq 0$. We have the following convergence theorems in which $V$ controls the convergence speed.

**Theorem 5.2.** *Under Assumption 5.1 and Assumption 4.1, let $\boldsymbol{\theta}(t)$ be the parameters of model (4) trained by Gradient Descent (2) starting from random initialization (5). Let the width $m = \Omega(\log(n/\delta))$, the initialization scale $\kappa = \mathcal{O}(\eta_0 \mu_0/n)$ in (5), $V$ be defined in (7), $c$ be a constant in $(0, \frac{1}{6(1+2\eta_0)^2+2}]$, the constant $c' > 0$ be sufficiently small, the hitting time $T_0 = \lceil (n \log 2)^{\frac{2}{Vc}}\rceil$, the parameter $r \in [1, +\infty)$ and the learning rate satisfy*

$$\begin{cases} \eta_0 \leq 1/2\sqrt{2}, & \text{for } t = 0 \\ \eta_t = \frac{c}{t\mathcal{L}(\boldsymbol{\theta}(t))}, & \text{for } 1 \leq t < T_0 \\ \eta_t = \frac{c'}{\mathcal{L}(\boldsymbol{\theta}(t))^{1-\frac{1}{2r}}}, & \text{for } t \geq T_0 \end{cases}.$$

*Then, with probability at least $1 - \delta - 2me^{-2d}$, GD will converge at **polynomial** rate:*

$$\begin{cases} \mathcal{L}(\boldsymbol{\theta}(t)) \leq \frac{\mathcal{L}(\boldsymbol{\theta}(1))}{t^{\frac{Vc}{2}}}, & 1 \leq t < T_0 \\ \mathcal{L}(\boldsymbol{\theta}(t)) = \mathcal{O}\big(\frac{1}{t^r}\big), & t \geq T_0 \end{cases}.$$

**Theorem 5.3.** *Under Assumption 5.1 and Assumption 4.1, let $\{\boldsymbol{b}_k(t)\}_{k \in [m]}$ be the input layer parameters of model (4), and we only train this layer by Gradient Descent (2) starting from random initialization (5). Let the width $m = \Omega(\log(n/\delta))$, the initialization scale $\kappa = \mathcal{O}(\eta_0 \mu_0/n)$ in (5), the constant $V$ be defined in (7), the constant $c \leq \frac{1}{2}$ and the learning rate satisfy*

$$\begin{cases} \eta_0 \leq 1/2\sqrt{2}, & \text{for } t = 0 \\ \eta_t = \frac{c}{\mathcal{L}(\boldsymbol{\theta}(t))}, & \text{for } t \geq 1 \end{cases}.$$

*Then with probability at least $1 - \delta - 2me^{-2d}$, GD will converge at **exponential** rate:*

$$\mathcal{L}(\boldsymbol{\theta}(t)) \leq \left(1 - \frac{Vc}{2}\right)^{t-1} \mathcal{L}(\boldsymbol{\theta}(1)), \quad t \geq 1.$$

Theorem 5.2 and 5.3 describe the whole training process starting from random initialization to convergence. If all parameters in the network are trained, we obtain the global convergence with *arbitrary polynomial* rate. On the other hand, if only the input layer parameters $\{\boldsymbol{b}_k\}_{k \in [m]}$ are trained, we can obtain *exponential* convergence rate. Fixing some layers of the neural network is a common practice for theoretical studies (Chatterji et al., 2021b). The results show that adaptively increasing learning rate helps GD achieve faster convergence. Similar idea has been explored in previous works such as (Lyu and Li, 2019; Chatterji et al., 2021b).

A similar convergence result was proven in (Phuong and Lampert, 2020). where the authors showed a global convergence result of two-layer neural networks trained by Gradient Flow (GF) with cross entropy loss and orthogonal separable data. The main difference between our results and analysis with that in (Phuong and Lampert, 2020) is that we study GD instead of GF. Due to the discretization, GD is more complicated than GF. For example, the balanced property of two layers ($\sum_{k \in [m]} |a_k(t)|^2 - \sum_{k \in [m]} \|\boldsymbol{b}_k(t)\|^2 \equiv 0, \ \forall t \geq 0$) holds for GF throughout the training, but it does not hold for GD. Dealing with GD requires new analysis techniques and finer analysis, such as the neuron partition results we derive. Another related work is (Lakshminarayanan and Singh, 2020). The neural partition approach in our article shares some similarity with the view of gating networks. In their work, the authors use gating networks to understand the role of depth in training deep ReLU networks. By comparison, we focus more on the characterization of training dynamics of two-layer ReLU networks.

## 6  Proof Sketch and Techniques

### 6.1  Main techniques in the proof of Theorem 4.2

Inspired by the empirical work (Ma et al., 2020), we study the nonlinear behavior of each neuron in the early stage of training. First, we define the hitting time $T$, and we will study the convergence in the early stage $0 \leq t \leq T$:

$$T := \sup\left\{t \in \mathbb{N} : \max_{i \in [n]} |f(\boldsymbol{x}_i; \boldsymbol{\theta}(s))| \leq 1, \ a_k(s)a_k(0) > 0, \ \forall k \in [m], \ \forall 0 \leq s \leq t+1\right\}.$$

#### 6.1.1  Neuron partition: fine-grained dynamical analysis of each neuron and each sample

During our proof, we conduct fine-grained analysis to the interaction of neurons and samples. Specifically, we characterize the impact of each sample on each neuron, and for each sample classify neurons into four categories according to their effect. We call this analysis "neuron partition".

For the $i$-th training data $(\boldsymbol{x}_i, y_i)$, we divide all neurons into four categories during training $0 \leq t \leq T$ and study them separately. We define the true-living neurons $\mathcal{TL}_i(t)$, the true-dead neurons $\mathcal{TD}_i(t)$, the false-living neurons $\mathcal{FL}_i(t)$ and the false-dead neurons $\mathcal{FD}_i(t)$ at time $t$ as:

$$\mathcal{TL}_i(t) := \left\{k \in [m] : y_i a_k(t) > 0, \boldsymbol{b}_k(t)^\top \boldsymbol{x}_i > 0\right\},$$
$$\mathcal{TD}_i(t) := \left\{k \in [m] : y_i a_k(t) > 0, \boldsymbol{b}_k(t)^\top \boldsymbol{x}_i \leq 0\right\},$$
$$\mathcal{FL}_i(t) := \left\{k \in [m] : y_i a_k(t) < 0, \boldsymbol{b}_k(t)^\top \boldsymbol{x}_i > 0\right\},$$
$$\mathcal{FD}_i(t) := \left\{k \in [m] : y_i a_k(t) < 0, \boldsymbol{b}_k(t)^\top \boldsymbol{x}_i \leq 0\right\}.$$

It is easy to verify $[m] = \mathcal{TL}_i(t) \bigcup \mathcal{TD}_i(t) \bigcup \mathcal{FL}_i(t) \bigcup \mathcal{FD}_i(t)$ (see Lemma A.1).

Under a weak assumption on the width of the network, we have the following results for the neuron partition at initialization:

**Lemma 6.1** (Informal statement of Lemma A.2). *If $m = \Omega(\log(n/\delta))$, then with high probability, if data $i$ and data $j$ are in the same class, we have card$\left(\mathcal{TL}_i(0) \cap \mathcal{TL}_j(0)\right) \approx \frac{\pi - \arccos(\boldsymbol{x}_i^\top \boldsymbol{x}_j)}{4\pi} m.$*

The next important lemma characterizes the evolution of the neuron partition under GD dynamics.

**Lemma 6.2** (Informal Lemma A.4). *With high probability, for any $i \in [n]$ and $t \leq T$ we have:*
*(S1) True-living neurons remain true-living: $\mathcal{TL}_i(t) \subset \mathcal{TL}_i(t+1)$.*
*(S2) False-dead neurons remain false-dead: $\mathcal{FD}_i(t) \subset \mathcal{FD}_i(t+1)$.*
*(S3) True-dead neurons turn true-living in the firt step ($t = 0$): $\mathcal{TD}_i(0) \subset \mathcal{TL}_i(1)$.*
*(S4) False-living neurons turn false-dead in the firt step ($t = 0$): $\mathcal{FL}_i(0) \subset \mathcal{FD}_i(1)$.*
*(S5) For any $i \in [n]$, $1 \leq t \leq T$, $k \in [m]$ and $\boldsymbol{b} \in \overline{\boldsymbol{b}_k(t)\boldsymbol{b}_k(t+1)}$, we have $\operatorname{sgn}(\boldsymbol{b}^\top \boldsymbol{x}_i) \equiv \operatorname{sgn}(\boldsymbol{b}_k^\top(1)\boldsymbol{x}_i) \neq 0$.*

To interpret Lemma 6.2, we display the first-step dynamics in Figure 1.

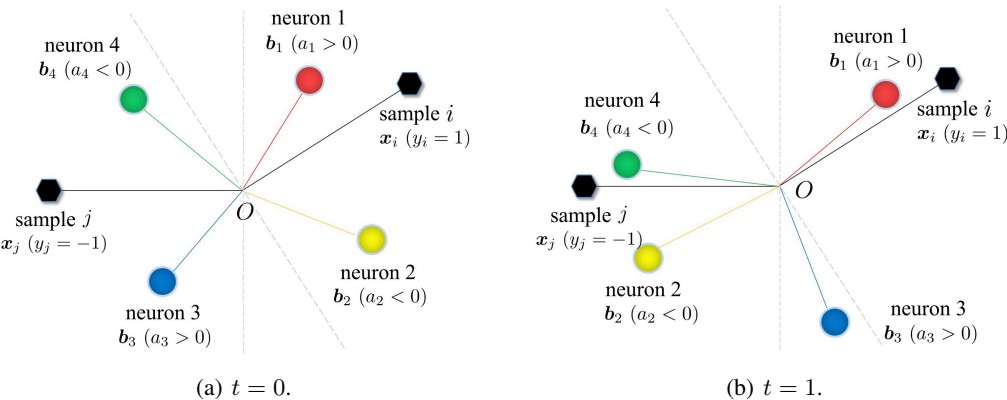

(a) $t = 0$.                    (b) $t = 1$.

Figure 1: The first-step dynamics of four neurons which belong to different neuron partitions at $t = 0$ for sample $i$ (with label 1) and sample $j$ (with label $-1$). Fig 1(a) and Fig 1(b) depict the directions of each neuron at $t = 0$ and $t = 1$, respectively. Results in Lemma 6.2 can be visualized: (S1) neuron $1 \in \mathcal{TL}_i(0)$ and $1 \in \mathcal{TL}_i(1)$. (S2) neuron $4 \in \mathcal{FD}_i(0)$ and $4 \in \mathcal{FD}_i(1)$. (S3) neuron $2 \in \mathcal{TL}_j(0)$ and $2 \in \mathcal{TL}_j(1)$. (S4) neuron $2 \in \mathcal{FL}_i(0)$ and $2 \in \mathcal{FD}_i(1)$. (Counterexample) By our Lemma, the first-step dynamics of neuron 3 cannot happen.

#### 6.1.2 Estimate of hitting time and parameters

Before the analysis of loss descent, we need to estimate the hitting time $T$ and the speed of change for each $a_k, \boldsymbol{b}_k$ in the early stage. Our tool is comparing the hitting time $T$ with an exponentially growing hitting time $T_e$ (see (12) in Appendix A).

$$T_e := \sup\left\{ t \in \mathbb{N} : \ (1 + 2\eta)^{2(t+1)} - (1 - 2\eta)^{2(t+1)} \lesssim 1; \ (1 + 2\eta)^{t+1} \lesssim 1 \right\},$$

where the order of magnitude of the two absolute constants hidden by $\lesssim$ is $10^0$.

Generally speaking, we can prove the following facts.

**Lemma 6.3** (Informal Lemma A.7 and A.8). *For any sample $i \in [n]$ and $t \leq T_e$, we have:*
*(P1) For any $k \in \mathcal{TL}_i(t)$, $a_k(t)\operatorname{sgn}(a_k(0))$ is non-decreasing, and $a_k(t)$, $\boldsymbol{b}_k(t)$ have an exponential upper bound.*
*(P2) For any $k \in [m]$, $a_k(t)$ has an exponential lower bound.*
*(P3) $f(\boldsymbol{x}_i; \boldsymbol{\theta}(t))$ has an exponential upper bound.*
*(P4) $T \geq T_e \geq T^* = \Theta(\frac{1}{\eta})$.*

#### 6.1.3 Gram matrix and gradient lower bound

The gradient lower bound is usually an important technical component in the analysis of loss descent. Our analysis establishes a novel lower bound for the gradient for mildly parameterized neural networks based on the neuron partition.

If we define the error for sample $i$ at $t$ as $e_i(t) = f(\boldsymbol{x}_i; \boldsymbol{\theta}(t)) - y_i$, and the Gram matrix at $t$ as $\mathbf{G}(t) = (\nabla f(\boldsymbol{x}_i; \boldsymbol{\theta}(t))^\top \nabla f(\boldsymbol{x}_j; \boldsymbol{\theta}(t)))_{(i,j) \in [n] \times [n]}$. For the analysis of GD, loss descent is

determined by $\|\nabla\mathcal{L}(\boldsymbol{\theta}(t))\|^2 = \sum_{i=1}^{n}\sum_{j=1}^{n} e_i(t)\mathbf{G}_{i,j}(t)e_j(t)$. Qualitatively, bigger gradient leads to the faster descent of the loss. To derive a tight lower bound for the gradient, we first derive the following bound for the entries of the Gram matrix:

**Lemma 6.4** (Informal Lemma A.9). *Under the condition of Theorem 4.2, with high probability, the Gram matrix* $\mathbf{G}(t) = \begin{pmatrix} \mathbf{G}_+(t) & \mathbf{0}_{\frac{n}{2}\times\frac{n}{2}} \\ \mathbf{0}_{\frac{n}{2}\times\frac{n}{2}} & \mathbf{G}_-(t) \end{pmatrix}$ *has the following form for any* $1 \leq t \leq T$ *and* $i, j \in [\frac{n}{2}]$:

$G_+(t)_{i,j},\ G_-(t)_{i,j} \gtrsim \boldsymbol{x}_i^\top\boldsymbol{x}_j\Big(\frac{\pi - \arccos(\boldsymbol{x}_i^\top\boldsymbol{x}_j)}{4\pi} - \mathcal{O}(\sqrt{\frac{\log(n/\delta)}{m}})\Big) > 0$, *where the order of magnitude of the absolute constant hidden by* $\gtrsim$ *is* $10^0$.

Combining the estimate above as well as estimates on parameters and the prediction function, we establish our gradient lower bound.

**Lemma 6.5** (Informal Lemma A.10). *For any* $1 \leq t \leq T^*$, $\|\nabla\mathcal{L}(\boldsymbol{\theta}(t))\| = \Omega\big(1 - (1 + 2\eta)^{2t}\big)$.

Besides, since we consider GD, the following per-step estimate of loss function (Bubeck et al., 2015) as well as the Hessian upper bound is also useful.

**Lemma 6.6** (Informal Lemma A.11 and A.12). *For* $1 \leq t \leq T^* - 1$, *we have* $H := \sup_{\boldsymbol{\theta}\ along\ GD\ trajectory} \big\|\nabla^2\mathcal{L}(\boldsymbol{\theta})\big\| = \mathcal{O}(1)$ *and* $\mathcal{L}(\boldsymbol{\theta}(t+1)) \leq \mathcal{L}(\boldsymbol{\theta}(t)) + \langle\nabla\mathcal{L}(\boldsymbol{\theta}(t)), \boldsymbol{\theta}(t+1) - \boldsymbol{\theta}(t)\rangle + \frac{1}{2}H\|\boldsymbol{\theta}(t+1) - \boldsymbol{\theta}(t)\|^2$.

## 6.2  Proof outline of Theorem 4.2

First, we can estimate the initial neuron partition (Lemma 6.1) at the initialization. Then, we study the dynamics of neuron partition trained by GD (Lemma 6.2), which gives a precise directional characterization about each neuron for each sample. Next, by comparing with an exponentially growing hitting time, we can estimate the hitting time $T$ and the parameters (Lemma 6.3). Combining Lemma 6.1, 6.2 and 6.3, we derive a dynamical lower bound of each element of Gram matrix (Lemma 6.4), which induces our gradient lower bound (Lemma 6.5). Finally, by combining the gradient lower bound (Lemma 6.5) and the loss upper bound (Lemma 6.6), we complete our proof. For more details, please refer to Appendix A.

The proof of Theorem 4.6 is similar to Theorem 4.2 and put into Appendix B.

## 6.3  Proof outline of Theorem 5.2 and 5.3

The proofs of Theorem 5.2 and 5.3 still rely on the dynamical analysis of neuron partition for each sample. First, the estimate of initial neuron partition (Lemma 6.1) also holds, and the dynamical analysis of neuron partition in Lemma 6.2 holds for any $t \geq 1$ (Lemma C.5 and C.6). Second, we can prove that the structure and lower bound of Gram matrix in Lemma 6.4 hold for any $t \geq 1$ (Lemma C.9), which induces the gradient lower bound below. Finally, combining the estimate of gradient lower bound, Hessian upper bound and loss upper bound, we complete our proof. For more details, please refer to Appendix C and D.

**Lemma 6.7** (Informal Lemma C.10). *Under the condition of Theorem 5.3, with high probability, we have the following lower bound for any* $t \geq 1$: $\|\nabla\mathcal{L}(\boldsymbol{\theta}(t))\| \gtrsim \mathcal{L}(\boldsymbol{\theta}(t))$, *where the absolute constant hidden by* $\gtrsim$ *is* $\sqrt{V}$, *defined in* (7).

In Lyu and Li (2019); Chatterji et al. (2021b), gradient lower bounds are derived for the late stage of training when $\mathcal{L}(\boldsymbol{\theta}(t)) < 1/n$, and take the form $\|\nabla\mathcal{L}(\boldsymbol{\theta}(t))\| \gtrsim \mathcal{L}(\boldsymbol{\theta}(t))\log\big(1/\mathcal{L}(\boldsymbol{\theta}(t))\big)/\|\boldsymbol{\theta}(t)\|$. Our lower bound, instead, works on the whole training process. With this stronger gradient lower bound, we can show the exponential convergence in Theorem 5.3.

Another important technical detail is the stability of GD, i.e., the learning rate should be upper bounded by the quantity related with the top eigenvalue of the Hessian. Our analysis also considered this factor, but implicitly. For global convergence results, the loss function we consider is exponential-type, whose Hessian can be controlled by the loss value. For example, for Theorem 5.2, there exists an absolute constant $C > 0$, s.t. $\big\|\nabla^2\mathcal{L}(\boldsymbol{\theta}(t))\big\| \leq Ct\mathcal{L}(\boldsymbol{\theta}(t))$ for any $t \geq 1$, please refer to Lemma C.15 in the appendix. Therefore, we can transform the Hessian-controlled learning rate condition into the loss-controlled learning rate condition in Theorem 5.2.

# 7 Experiments

**MNIST and CIFAR-10 experiments.** As we mentioned in Section 4.2, our theory about early stage convergence applies to a wide range of dataset (Assumption 4.5) such as MNIST and CIFAR-10. We use the two datasets (with normalization) to compare the experimental result with Theorem 4.6. Specifically, we use the first 1000 data in MNIST dataset and the first 1000 data in CIFAR-10 dataset, separately. (The two datasets with normalization $\|\boldsymbol{x}\| = 1$ both satisfy Assumption 4.5.) And we use the two-layer ReLU network with the logistic loss $\ell(\boldsymbol{y}_1, \boldsymbol{y}_2) = \log(1 + \exp(-\boldsymbol{y}_1^\top \boldsymbol{y}_2))$. $m, \kappa, \eta$ are choosen by Theorem 4.6. We study the change of our bounds under different network width $m$ and learning rate $\eta$. All experiments are conducted on a MacBook pro 13" only using CPU. See the code at `https://github.com/wmz9/Early_Stage_Convergence_NeurIPS2022`.

Table 1: Results of MNIST and CIFAR-10 experiments. In the first table, we fix $\eta = 0.01$ and change $m$; in the second one, we fix $m = 200$ and change $\eta$.

| $m$ | 100 | 200 | 500 | 1000 |
|---|---|---|---|---|
| Our hitting iteration | | 34 | | |
| Realistic loss descent (CIFAR-10) | $3.45 \times 10^{-1}$ | $5.34 \times 10^{-1}$ | $6.11 \times 10^{-1}$ | $6.53 \times 10^{-1}$ |
| Realistic loss descent (MINST) | $3.89 \times 10^{-1}$ | $4.78 \times 10^{-1}$ | $5.97 \times 10^{-1}$ | $6.45 \times 10^{-1}$ |
| Our loss descent bound | | $2.63 \times 10^{-1}$ | | |
| $\eta$ | 0.01 | 0.005 | 0.002 | 0.001 |
| Our hitting iteration | 34 | 69 | 173 | 346 |
| Realistic loss descent (CIFAR-10) | $4.89 \times 10^{-1}$ | $5.34 \times 10^{-1}$ | $5.43 \times 10^{-1}$ | $5.37 \times 10^{-1}$ |
| Realistic loss descent (MINST) | $4.78 \times 10^{-1}$ | $4.98 \times 10^{-1}$ | $4.95 \times 10^{-1}$ | $4.92 \times 10^{-1}$ |
| Our loss descent bound | | $2.63 \times 10^{-1}$ | | |

The results in Table 7 show that our loss descent bound (5-th row in each table) is **relatively tight**, basically in the same order of magnitude as the realistic loss descent (3-rd row and 4-th row in each table). And our loss descent bound does not change with the choice of different datasets.

# 8 Conclusion and Future Work

In this paper, we study the convergence of GD and SGD when training mildly parameterized neural networks. On one hand, we show early stage convergence for a wide range of loss functions, optimization algorithms, and training data distributions. On the other hand, under some assumptions on the loss function and data distribution, we show global convergence of GD. Our analysis can be extended to the minimization of population risk (See Appendix E).

The theoretical understanding of the training of neural networks, especially for neural networks with practical sizes, still has a long way to go. For instance, although the late stage training for exponential loss function is simple and clear, that for other losses is much more complicated. Phenomena like unstable convergence (Wu et al., 2018; Cohen et al., 2021; Ahn et al., 2022; Ma et al., 2022) happen. Better understanding of these phenomena during training is an important direction of future work.

## Acknowledgments and Disclosure of Funding

We thank anonymous reviewers for helpful suggestions. Mingze Wang is supported in part by the National Key Basic Research Program of China: 2015CB856000.

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
