# OpenReview forum: "Early Stage Convergence and Global Convergence of Training Mildly Parameterized Neural Networks"
_NeurIPS.cc/2022/Conference — NeurIPS 2022 Accept_

### Official Review · Reviewer_8f7C · 2022-07-04

**Rating:** 7
**Confidence:** 5
**Soundness:** 3 good
**Presentation:** 3 good
**Contribution:** 3 good

**Summary:**

This paper studies the convergence of GD and SGD when training mildly parameterized neural networks starting from random initialization. The authors prove an "early stage convergence" that the loss is decreased by a significant amount in the early stage of the training quickly.

**Questions:**

Assumption 4.1 assumes that close data points have same output, this also implies the training data has significant low-frequency components. According the frequency principle (https://arxiv.org/abs/1901.06523), low frequency would converge at the beginning and fast, Theorem 4.2 can be intuitively understood. For Assumption 4.5, it constraints the input but not the output. That means Theorem 4.6 applies for any training data. However, in practical training, if the training data is dominated by very high frequency components, the loss may not decay or decay extremely slow. Does this example show that the \Omega(1) bound in theorem 4.6 may be loose?

The model in Theorem 4.2 lacks of universal approximation ability (without bias), how does this result help understand a model with bias.

A loss that satisfies Assumption 4.4 may not a common case in deep learning, such as MSE and cross-entropy. Can authors explain more about how Theorem 4.6 depends on the Assumption 4.4?

I am still confusing about how Theorem 4.2 and 4.6 help understanding the training process. For example, can these two theorems tell us in which situation there is an early-stage convergence and there is not, or to achieve a fast convergence at the initial stage, how should we design the network.

Regarding the global convergence part, the results heavily relay on the Assumption 4.1 and model (4), which seems not very realistic. Can authors discuss more on how these results can help understand the practical neural network?


**Limitations:**

The theoretical results in this paper are nice and sound solid, however, I suggest the author should make the limitation clear in abstract and introduction, which are now quite misleading. The informal statements of theorem ignore very important assumptions, which will mislead readers by regarding the results as very general results. The title and the abstract should explicitly point out the paper studies two-layer neural network at least. These limitations at least include the model, loss and data.

**Strengths And Weaknesses:**

The paper is well-written, technique sound, and studies important problems of neural network training process, one is the fast convergence at the early training stage, the other is the global convergence of mildly-overparameterized neural networks. However, some assumptions may be not so realistic, which makes reader hard to connect the results and practical training process.

---

> ### Author Response · Authors · 2022-08-02
> **Response to Reviewer 8f7C (Part II)**
>
> **Q4. I am still confused about how Theorem 4.2 and 4.6 help understanding the training process. For example, can these two theorems tell us in which situation there is an early-stage convergence and there is not, or to achieve a fast convergence at the initial stage, how should we design the network.**
>
> **Response:** By our theorem, the early stage convergence always happens as long as the assumptions are satisfied. These results theoretically consolidate the common sense that neural network losses trained by GD decreases fast at the beginning of the training. The length of the early-stage-training period and its connection with generalization performance is an important direction of future work. Our estimate for the ``hitting iteration'' for simple cases can serve as a starting point. Refining the analysis for estimating the hitting iteration may reveal the early-stage training behavior for different network architectures, dataset, and even optimization algorithms.
>
>
> **Q5. Regarding the global convergence part, the results heavily rely on the Assumption 4.1 and model (4), which seems not very realistic. Can authors discuss more on how these results can help understand the practical neural network?**
>
> **Response:**  The assumptions and restrictive form of the model is employed to produce rigorous and comprehensive theoretical results. Without the assumptions, strict theorems may not hold, but similar behavior may still exists. Some examples are shown in our numerical experiments (please refer to the response to reviewer nRXr). Also, it is possible to relax some of the assumptions in the theory by finer analysis. For instance, bias can be considered in model (4) as shown by Theorem 4.6, which makes the model a little more realistic and satisfy the universal approximation property.
>
> **Q6. About the Limitations.**
>
> **Response:** We sincerely apologize for the misleading caused by limitations. In the revised version of the paper, we will make more precise statements, including citing data assumptions in the two informal theorems and adding more precise restrictions in our title and abstract.

---

> ### Author Response · Authors · 2022-08-02
> **Response to Reviewer 8f7C (Part I)**
>
> We thank the reviewer for appreciating our work, as well as the valuable suggestions and comments. We address the questions in the following.
>
> **Q1. Assumption 4.1 assumes that close data points have same output, this also implies the training data has significant low-frequency components. According the frequency principle [1], low frequency would converge at the beginning and fast, Theorem 4.2 can be intuitively understood. For Assumption 4.5, it constraints the input but not the output. That means Theorem 4.6 applies for any training data. However, in practical training, if the training data is dominated by very high frequency components, the loss may not decay or decay extremely slow. Does this example show that the $\Omega(1)$ bound in theorem 4.6 may be loose?**
>
> **Response:** Even if the training data has high frequency, there will be a low frequency component (actually a dc-component) because the label vectors we consider (in Section 4.2) are standard unit vectors. In the early stage of the training, this low frequency component is learned fast, and our results characterize this process. The related bounds in Theorem 4.6 are relatively tight, please refer to the first numerical experiment in our response to reviewer nRXr.
>
> **Q2.The model in Theorem 4.2 lacks of universal approximation ability (without bias), how does this result help understand a model with bias.**
>
> **Response:** Actually, our results on the early stage convergence hold for ReLU networks with or without bias. For the binary classification problem (Theorem 4.2), we presented results without bias because in this setting we have a sharper loss descent bound.
>
> **Q3. A loss that satisfies Assumption 4.4 may not a common case in deep learning, such as MSE and cross-entropy. Can authors explain more about how Theorem 4.6 depends on the Assumption 4.4?**
>
> **Response:** Though the quadratic loss is not included, many widely used loss functions satisfy Assumption 4.4, such as hinge loss, logistic loss and exponential loss. For loss functions that satisfies the assumption, the only term containing $\boldsymbol{f}(\boldsymbol{\theta})$ is $\boldsymbol{y}^\top\boldsymbol{f}(\boldsymbol{\theta})$, hence we can easily compute and estimate the gradient and the Hessian according to the chain rule. For the quadratic loss, the gradient and Hessian still have a relatively neat form. To obtain a similar theorem for multi-class problems, we only need to extend Theorem 4.2 from 1-dimension to $c$-dimension. In the revised paper, we will add a proof for the quadratic loss in the multi-class setting. The analysis of the cross-entropy loss in the multi-class setting might be more involved. We will leave it for future work.
>
>
> &nbsp;
> &nbsp;
> ----
>
> **Reference**
>
> [1] Xu, Zhi-Qin John, et al. Frequency principle: Fourier analysis sheds light on deep neural networks. arXiv preprint arXiv:1901.06523, 2019.

---

### Official Review · Reviewer_uRDi · 2022-07-07

**Rating:** 5
**Confidence:** 4
**Soundness:** 3 good
**Presentation:** 3 good
**Contribution:** 2 fair

**Summary:**

This paper studies the early stage of convergence for moderate width networks optimized via the squared loss and logistic-style loss functions.  More precisely, they demonstrate that the loss will descend by an $\Omega(1)$ quantity in the early stage of training.  If the data are very well-separated, for logistic-style loss functions they can demonstrate global convergence.  A contribution is that they only require logarithmic width for the network.  As a matter of technique the novelty is that rather than prove NTK deviation bounds they control the NTK gram matrix via fine grained analysis of the neurons.  While previous works such as [1] and [2] use overparameterization to demonstrate that the activation patterns do not change much, the work under consideration does not rely on overparameterization and performs a more refined analysis of neurons based on true/false-alive/dead pairings.

References:

[1] Gradient Descent Provably Optimizes Over-parameterized Neural Networks by Du et al.

[2] Fine-Grained Analysis of Optimization and Generalization for Overparameterized Two-Layer Neural Networks by Arora et al.

**Questions:**

Most of my questions were outlined in the “Strengths and Weaknesses” section, however I will reiterate these here as well as introduce some new questions.

Question 1: Could the authors provide a more detailed discussion of how their analysis of the early stage of training is able to improve over what more standard NTK analysis would give.

Question 2: 2nd inequality of line 570 could be explained better.

Question 3: In Lemma A.13 the authors upper bound the gradient deviations using the loss Hessian.  However, for ReLU the loss Hessian does not exist in the proper sense.  I believe this can be overcome using a different argument however this discrepancy must be addressed or at least explained.

Question 4: Could the authors provide a more detailed discussion of the scale of the initialization.  If $\kappa$ is small then the gradients of the inner layer weights will dominate over the gradients of the outer layer weights (due to properties of ReLU).  This is not bad necessarily however this should be explained to the reader.  I believe this is essential to the proofs.  For example the authors study the stopping time $T$ up to which the signs of the outer layer weights haven’t changed.  The size of $\kappa$ will be quite relevant for determining how quickly the outer layer weights change in sign.

Question 5: Could the authors provide a more detailed comparison of their work and [3] which addresses some of my concerns outlined in the “Strengths and Weaknesses” section.

References:


[3] Polylogarithmic width suffices for gradient descent to achieve arbitrarily small test error with shallow ReLU networks by Ji and Telgarsky https://arxiv.org/pdf/1909.12292.pdf

**Limitations:**

The main contribution of this work as I see it is to provide a fine grained analysis of the neurons for the shallow ReLU network.  This is indeed interesting, although it will be difficult to generalize to multiple layers or to activation functions other than ReLU.

Based on my concerns in the "Strengths and Weaknesses" section, I do not view the present version of this manuscript to be ready for publication in this conference.  If the authors provide strong responses to my concerns to show that I am mistaken or missed crucial points I will consider raising my recommendation.

**Edit after rebuttal: I upgraded my recommendation from a 3 to a 5 based on the edits and responses from the author.  Please see my comments below for my reasoning.**


**Strengths And Weaknesses:**


The strength of the paper is that it only requires logarithmically wide networks and it provides a fine-grained analysis of the neurons that is novel as far as I can tell.  Furthermore the writing and presentation is easy to follow.  However, I have some concerns about this work.  I would like the authors to provide a more detailed explanation about how their analysis is able to prove results that cannot be proven using existing tools from NTK theory. If one is only interested in analyzing the early stage of training one does not need overparameterization to perform standard NTK analysis.  Up to a $O(1)$ stopping time, parameter deviations are $O(1)$ and thus NTK deviations are $O(1/\sqrt{m})$.  Thus if one is only interested in the early-time dynamics the width of the network can be taken independent of $n$.  As far as the statements for global convergence, under Assumption 4.1 the data are linearly separable.  When the data are linearly separable with positive margin, the target function is separable with positive margin using the NTK model see [3].  For this case [3] demonstrated that polylogarithmic width suffices for the logistic loss to achieve global convergence.  To be fair in [3] they only train the hidden layer, however due to $\kappa$ being small in the present work the gradients of the hidden layer will typically be larger than the gradients for the outer layer.  In this case the changes in the hidden layer should be more significant than the changes in the outer layer.  Thus I do not see this being too essential here.  Also in Theorem 5.3 the authors only train the hidden layer which allows a more direct comparison.

References:

[3] Polylogarithmic width suffices for gradient descent to achieve arbitrarily small test error with shallow ReLU networks by Ji and Telgarsky https://arxiv.org/pdf/1909.12292.pdf

---

> ### Author Response · Authors · 2022-08-02
> **Response to Reviewer uRDi (Part II): comparison of our work and [5]**
>
> **Q5. Could the authors provide a more detailed comparison of their work and [5] which addresses some of my concerns outlined in the “Strengths and Weaknesses” section.**
>
> **Response:** We thank the reviewer for pointing out the reference. We carefully compare our work with [5] in the following aspects. A related discussion will be added to the revised version of the paper.
> - **(1) Training dynamics.** The training dynamics is one of the main differences. It is well known that NTK describes a lazy training process, i.e., the parameters keep close to the initialization throughout the training process, and the neural network's dynamics is close to that of the kernel method. However, our paper portrays richer training dynamics beyond lazy training. In short, the initialization neighborhood of 0 is a saddle region for Gradient Descent, and we demonstrate a two-stage **``saddle escaping $\to$ convergence'' dynamics** that does not require the parameters to stay near the initialization (Theorem 5.2 and 5.3). Specifically, we show that: firstly, neurons will adjust their directions rapidly and escape the saddle region in the **first** step, and training accuracy will rise to 100\%; then, GD will enter a good region that contains neither spurious local minima nor saddle points, and neurons will keep moving through the right directions until convergence. Besides, in our analysis, neurons can go far rather than keep close to the initialization.
>
> - **(2) The role of parameterization.** In [5], the main role of over-paramterization is to ensure that the initialization has a positive margin (Lemma 2.3 in [5]), and parameters will keep close to the initial positions (equ(A.8) in [5]) and keep a good margin during training. However, we **do not need** the initialization has a good margin. We only need $m=\Omega(\log(n/\delta))$ to ensure that the intersection of initial set $\mathcal{TL}_i(0)$ and $\mathcal{TL}_j(0)$ is not empty. Under our mildly parameterization, the initialization may not have a positive margin, but after just one step of GD, neural networks will have a 100\% training accuracy (Lemma C.7 in our work). Besides, we do not need to resort to over-parameterization like NTK to ensure that the parameters keep close to the initialization during training, which is why the neurons can go far in our mildly parameterized regime.
>
> - **(3) The scale of parameterization.** From the discussion in (b), the role of parameterization in our work is not as significant as in [5], which can also be verified by comparing the initialization scale in the two works. Our work only needs $m=\Omega(\log(n/\delta))$, but their work need $m=\Omega(\frac{\text{poly}(\log(n/\delta))}{\gamma^8})$ (Theorem 2.2 in [5]). Furthermore, they have proved that $m=\Omega(\text{poly}(\frac{1}{\gamma}))$ is a lower bound of width to ensure initial separability (Page 10 in [5]), which is necessary for their NTK analysis. The parameter $\gamma$ is defined in the assumption 2.1 in [5]. As discussed in Section 5.1 in [5], for linearly separable data, $\gamma$ may be dependent on the margin $\gamma_0$, i.e., $\gamma=\gamma_0/2$. So if $\gamma_0$ is dependent on the input dimension $d$, their result will need a large $m=\Omega(\text{poly}(d\log(n/\delta)))$ for **high-dimensional** problems, however, our results are almost independent of the input dimension $d$. *For example, we can consider the following $2d$ training data ($n=2d$): For $i\in[d]$, $x_i=e_i\in\mathbb{R}^d$ with $y_i=1$; $x_{d+i}=-e_i\in\mathbb{R}^d$ with $y_{d+i}=-1$, where $e_i$ is $i-$th coordinate base in $\mathbb{R}^d$, i.e. $e_{ii}=1$ and $e_{ij}=0$ for $j\ne i$. It is easy to verify $\gamma_0=1/\sqrt{d}$ and this data distribution satisfies to our Assumption 4.1. For this distribution, their results need $m=\Omega(d^4\text{poly}(\log(n/\delta)))=\Omega(d^4\text{poly}(\log(d/\delta)))=\Omega(n^4\text{poly}(\log(n/\delta)))$, but ours only need $m=\Omega(\log(n/\delta))=\Omega(\log(d/\delta))$.* Hence, the scale $m$ of parameterization between our work and [5] is different.
>
> &nbsp;
> &nbsp;
> ----
> **Reference**
>
> [5] Ji, Ziwei and Telgarsky, Matus. Polylogarithmic width suffices for gradient descent to achieve
> arbitrarily small test error with shallow relu networks. arXiv preprint arXiv:1909.12292, 2019

---

> ### Author Response · Authors · 2022-08-02
> **Response to Reviewer uRDi (Part I)**
>
> We thank the reviewer for the comments and questions, which we address in the following.
>
> **Q1. Could the authors provide a more detailed discussion of how their analysis of the early stage of training is able to improve over what more standard NTK analysis would give.**
>
> **Response:** For the early stage convergence, our analysis gives the following results (Theorem 4.2 and 4.6): If we want the loss descent to be $\Omega(1)$, the time needed is $\Theta(\frac{1}{\eta})$; but if we use standard NTK analysis [1][2], the time needed is $\Theta(\frac{1}{\eta\lambda_{\min}(\mathbf{H}^{\infty})})$ (to ensure the loss descent is $\Omega(1)$).
> Here, $\mathbf{H}^{\infty}$ is the Gram matrix of NTK [2]. However, $\lambda_{\min}(\mathbf{H}^{\infty})$ is usually small [3]. Hence, our analysis treats a case in which is convergence is faster than NTK. Besides, our requirement on the learning rate, $\eta=O(1)$,  is also weaker than that in the NTK analysis, e.g. $\eta=O(\frac{\lambda_{\min}(\mathbf{H}^{\infty})}{n^2})$ in [2].
>
> **Q2. 2nd inequality of line 570 could be explained better.**
>
> **Response:** For simplicity, we denote $I_j:=\mathbb{I}${$\boldsymbol{b}_k(0)^\top\boldsymbol{x}_j\geq0$}$((y_j-f(\boldsymbol{x}_j;\boldsymbol{\theta}(0)))$
> $\boldsymbol{x}_j^\top\boldsymbol{x}_iy_i)(y_ia_k(0))$ in 2nd inequality of line 570.
> From $|f(\boldsymbol{x}_j;\boldsymbol{\theta}(0))|\leq2\kappa<1$, we know $\text{sgn}(y_j-f(\boldsymbol{x}_j;\boldsymbol{\theta}(0)))=\text{sgn}(y_j)$. Recalling Assumption 4.1, we have $(y_j-f(\boldsymbol{x}_j;\boldsymbol{\theta}(0)))\boldsymbol{x}_j^\top\boldsymbol{x}_i y_i\geq0$ for any $i,j\in[n]$. Then combining $y_i>0$ and $a_k(0)<0$, we know that $I_j\leq0$ for any $j\in[n]$.
>
> So we have $\sum_{j\in[n]}$$I_j=I_i+\sum_{j \ne i}I_j \leq I_i$, which means 2nd inequality holds.
>
>
> **Q3. In Lemma A.13 the authors upper bound the gradient deviations using the loss Hessian. However, for ReLU the loss Hessian does not exist in the proper sense. I believe this can be overcome using a different argument however this discrepancy must be addressed or at least explained.**
>
> **Response:** We apologize for not clearly discussing this issue in the main text. At the end of Page25 in the appendix, we have a note that discusses the ReLU problem. Although ReLU is not differentiable at $0$, we just define $\sigma''(0)=0$. Moreover, a more rigorous substitute is the high-order smooth ReLU, such as Huberized ReLU in some previous work [4]. For the high-order smooth ReLU, the 2nd derivative or 2nd sub-derivative can be defined at $0$. It is worth mentioning that our analysis also holds for the smooth case, the only difference is the absolute constant in our Hessian upper bounds. And similar Hessian estimate with Huberized ReLU can be found in Lemma 6 in [4].
>
>
> **Q4. Could the authors provide a more detailed discussion of the scale of the initialization. If $\kappa$ is small then the gradients of the inner layer weights will dominate over the gradients of the outer layer weights (due to properties of ReLU). This is not bad necessarily however this should be explained to the reader. I believe this is essential to the proofs. For example the authors study the stopping time $T$ up to which the signs of the outer layer weights haven’t changed. The size of $\kappa$ will be quite relevant for determining how quickly the outer layer weights change in sign.**
>
> **Response:** The initialization scale is indeed essential in our analysis. In fact, as the reviewer states, the small initialization of the first layer can guarantee a significant separation between the evolution speed of different layers at the early stage, which is often called a ``speed separation''. At the early stage, the weights of the second layer change very slowly, and the sign of the second layer does not change, i.e., neurons do not change from predicting correctly back to predicting incorrectly, which is important in our proof. In the revised version of the paper, we will add a discussion about the role played by the initialization scales.
>
>
>
>
>
> &nbsp;
> &nbsp;
> ----
>
> **Reference**
>
> [1] Du, Simon S, et al. Gradient descent provably optimizes over-parameterized neural networks. arXiv preprint arXiv:1810.02054, 2018.
>
> [2] Arora, Sanjeev, et al. Fine-grained analysis of optimization and generalization for overparameterized two-layer neural networks. International Conference on Machine Learning, 2019.
>
> [3] Xie, Bo, et al. Diverse neural network learns true target functions. Artificial Intelligence and Statistics, 2017.
>
> [4] Chatterji, Niladri S, et al. When does gradient descent with logistic loss find interpolating two-layer networks? J. Mach. Learn. Res., 2021.

---

> > ### Comment · Reviewer_uRDi · 2022-08-05
> > **Response to initial author comment (part 1)**
> >
> > Thanks for addressing my questions.  I list my responses to your comments on each question below.
> >
> > Q1:
> >
> > The time to convergence needed for worst case NTK analysis will indeed be $\Omega([\log(\frac{1}{1 - \frac{\eta}{n} \lambda_{min}(H^\infty)})]^{-1})$  (note one must divide the learning rate by $n$ if working with mean squared error).    However this worst case analysis corresponds to taking an estimate where you assume the residual (the difference between the network predictions and labels) lies in the bottom eigenspace of the NTK.  However, this is not a fair comparison in this case as you are assuming the data come from a very structured target function (members of the same/different class have positive/negative cosine).  In this case as you show in Lemma A.11 you have a stronger gradient lower bound than the estimate one gets if one assumes the residual lies in the bottom eigenspace.  Also note that while Arora et al. [1] require the learning rate is $\eta = O(\lambda_{min}(H^\infty) / n^2)$, this is not necessary in general for NTK analysis.  For the mean squared error for training to be stable you need that the matrix $(I - \frac{\eta}{n} H^\infty)$ has all its eigenvalues in $(-1, 1)$ which holds as long as $\eta < \frac{2 n}{\lambda_{max}}$.  The critical learning rate for the mean squared error would be to have $\eta < \frac{2n}{\lambda_{max}}$.  Since $\lambda_{max} / n = O(1)$, this means that you can set the learning rate $\eta = \Omega(1)$.  Also see [4] where they use the slightly different critical learning rate $\eta_{critical} := \frac{2}{\lambda_{min}(H^\infty) + \lambda_{max}(H^\infty)}$ which after translating to the mean squared error gives $\eta_{critical} := \frac{2n}{\lambda_{min}(H^\infty) + \lambda_{max}(H^\infty)}$.  In this case same argument holds, you can set $\eta = \Omega(1)$.
> >
> > Q2:
> > Thank you for the clarification.  It may be worth it to add the explanation to the manuscript.
> >
> > Q3:
> > To clarify, I did notice the footnote during the first read through.  I meant to say that this footnote is not adequate justification.  You are performing the proof as if the loss gradient is Lipschitz (with Lipschitz constant upper bounded by the operator norm of the Hessian).  However for ReLU the loss gradient is *not* Lipschitz (the problem is non-smooth).  In fact, this is one of the fundamental problems that one must overcome when performing convergence analysis for ReLU.  The number of activation patterns that change can be taken as a measure of how non-smooth the ReLU problem is.  The way prior works [1],[2] overcome the non-smoothness problem is that they upper bound the number of activation patterns that change.  The non-smoothness of the ReLU problem is a highly nontrivial technical obstacle.  This footnote is not adequate justification of your proof.  Unfortunately your current proof does not go through for the ReLU activation function.  Until this detail is addressed the theorems should not be stated for the ReLU activation function.  If the proofs are written for the smooth case then the statements of the theorems should reflect that.  I will not upgrade my recommendation until the manuscript addresses this discrepancy (by either fixing the proof or changing the statements of the theorem).  I am even more concerned over this detail than when I wrote my initial review.
> >
> > Q4: Thank you for explaining how the scale of the initialization is highly relevant for the proof.  I do think it is highly important for this to be explained to the reader, and look forward to the new version of the manuscript that addresses this.
> >
> > Q5: Thank you for the comparison.  I do agree that your analysis is different from [3] and thus you will have different requirements.  I do think that there is also value to providing different proof techniques aside from having different final bounds.  I do think it is important to compare against this work in the manuscript as some readers may want to see an explanation of these differences.

---

> > > ### Author Response · Authors · 2022-08-07
> > > **Response to Reviewer uRDi’s updated comments (Part II)**
> > >
> > > **2. Further Response to Q3**
> > >
> > > **Response:**  We thank the reviewer for pointing out the problems in our original approach when dealing with ReLUs. We agree that a local smoothing is not sufficient to include ReLU activation into the theory, because of the unbounded Lipschitz constant issue of the gradient. Inspired by the valuable idea provided by the reviewer, we **have fixed this issue** rigorously in our revised version. Please take a look at the new file named **"Appendix\_for\_Q3\_about\_smoothness.pdf"** in "Supplementary Material". In this file, we highlighted the changes we made in red. To summarize, we have made a more fine-grained analysis to address this issue.
> > >
> > > In the following, we take Theorem 4.2 and Appendix A as examples to explain our approach.
> > >
> > > * As a technical requirement, we add Assumption 4.1 (ii) (Line 130) about the data distribution. Fortunately, Assumption 4.1 (ii) is not a strong addition over Assumption 4.1 (i). For example, if there exists a $(x_0,y_0)$ in the dataset $S$ that satisfies $(-x_0,-y_0)$ also belongs to $S$, then the dataset $S$ satisfies Assumption 4.1 (ii) with a big $\mu_0=1$ (the upper bound of $\mu_0$ is $1$).
> > >
> > > * With the help of the technical assumption, we can make a fine-grained characterization of the training dynamics. By choosing initialization scale $\kappa=O(\eta\mu_0/n)$, we can prove that the transformation of all neuron partitions will be completed in the first step (from $t=0$ to $t=1$) in Lemma 6.2 (S3)(S4) (Line 273). Then from $t=1$ till the hitting time $t=T$, as stated in Lemma 6.2 (S5) (Line 275), neuron partitions no longer change along the Gradient Descent trajectory, and each neuron $b_k$ in the GD trajectory satisfies $\text{sgn}(b_k^\top x_i)\equiv\text{sgn}(b_k(1)^\top x_i)\ne 0$, which means $b_k^\top x_i\not\equiv0$. (Please refer to Lemma A.2 in Appendix for proof details.)
> > >
> > > * By Lemma 6.2 (S5), we know that after $t=1$, any parameter $\theta=(a_1,\cdots,a_m,b_1^\top,\cdots,b_m^\top)^\top$ along the GD trajectory satisfies $b_k^\top x_i\not\equiv0$ for any $k\in[m]$ and $i\in[n]$. It means $\text{ReLU}''(b_k^\top x_i)\equiv0$ holds strictly and the loss is smooth after $t=1$. Hence, in Lemma A.12 (line 668), we can derive the correct Hessian upper bound along GD trajectory after $t=1$. Then we can use the loss descent estimate for smooth loss (Lemma A.13) during $t=1$ to $t=T^*$ to derive the loss descent $L(\theta(1))-L(\theta(T^*))=\Omega(1)$ (Line 713).
> > >
> > > * The last step is to estimate the loss deviation from $t=0$ to $t=1$. Although the loss may be non-smooth in the first step, we can roughly estimate that the loss increase is no more than $O(\eta)$ by the first-order Taylor expression in Lemma 15 (Line 706). Finally, we obtain our loss descent $L(\theta(0))-L(\theta(T^*))=(L(\theta(1))-L(\theta(T^*)))+(L(\theta(0))-L(\theta(1)))=\Omega(1)-O(\eta)=\Omega(1)$.
> > >
> > > The correction of the proof can be similarly applied to the analysis of the full global convergence results. We only need to generalize the Hessian analysis to any $t\geq1$. In our new file ``Appendix\_for\_Q3\_about\_smoothness.pdf'', we also changed both Theorem 5.2 and Appendix C. The analysis of Theorem 4.6 and 5.2 can be changed in the same way as Theorem 4.2 and 5.3, respectively. We will make the changes in the final version. Finally, we remark that these changes of analysis and the additional assumption are for nonsmooth activation functions like ReLU. Our original analysis and assumptions still work perfectly on smooth activation functions.

---

> > > > ### Comment · Reviewer_uRDi · 2022-08-08
> > > > **Response part 2**
> > > >
> > > > Thank you for the corrections.  I am glad we were able to identify this issue before the deadline.  I do not see any technical problem with the corrections.  I will admit that Assumption 4.1 (ii) is not so intuitive and I do think it is still fairly strong to assume each data point has another data point pointing in the exact opposite direction (even though I know that is a stronger condition than Assumption 4.1 (ii)).  Please consider elaborating on this assumption in the final version.  My current recommendation is split between a 4 and a 5.  I still have some reservations over the significance of the loss descending by an $\Omega(1)$ quantity at the beginning of training.  Furthermore, I am a bit concerned over the important technical detail of the non Lipschitz gradient only being identified and fixed at the last moment.  Nevertheless the neuron partition is a novel approach to the problem and due to this fact and the fact that the authors did extensive work to prove multiple statements including the global convergence result I am  giving the authors the benefit of the doubt and raising my recommendation to a 5.  I strongly encourage the authors to try to take the feedback from the other reviewers (as well as my own) and to clean up the draft as much as possible to be ready for the final version.

---

> > > > > ### Author Response · Authors · 2022-08-09
> > > > > **Will keep improving the manuscript**
> > > > >
> > > > > We thank the reviewer for the thoughtful decision and the suggestions! We have made a first try to address the suggestions from all reviewers in the draft. The result is a revised version of the paper, uploaded into the "supplementary material" section. In the revised paper, we made careful changes on the old version according the the reviews. Major changes are marked red or orange. We thank the precious suggestions and comments from all the reviewers. More works need to be done to address all points in the reviews. We will keep improving the writing/structure/contents of the paper if we have the chance to submit later versions.

---

> > > ### Author Response · Authors · 2022-08-07
> > > **Response to Reviewer uRDi’s updated comments (Part I)**
> > >
> > > **General remark:** We agree with the reviewer that the behaviors that are studied in our work are in the NTK regime, namely the features caught by the network are not changing drastically during the training. Though, we would also like to highlight several major differences of our work from previous works on NTK:
> > >
> > > (1). We analyze the NTK of trained parameters, rather than initial random parameters. In our work, the feature representation is not fixed since initialization. Instead, the feature is learned very quickly from the data, and later get fixed. Therefore, compared with the purely linear dynamics in traditional NTK settings, in our setting the network experiences a nonlinear period before falling into a lazy regime. This is possible because the network is mildly parameterized---it does not fall into the lazy regime at the initialization.
> > >
> > > (2). Other than highly over-parameterization, we reveal a new setting in which lazy training may occur. In our setting, the NTK behavior is caused by the simplicity of training data, instead of over-parameterization. Because the training data have some structures, only a finite period of time is needed to learn the feature, leading the later dynamics to be linear. Our analysis shows that NTK is not restricted only in unrealistically over-parameterized models, but also relevant for practical-size models when the training data have some structures (which is usually the case for practical problems).
> > >
> > > Therefore, acting as a complementary of the large volume of previous works on NTK, our work deepens the understanding on the cause of NTK regime, and broadens the applicability of the NTK analysis.
> > >
> > > **For the reviewer's questions:** We appreciate the reviewer's affirmation of our response of Q2, Q4, and Q5. We will add the discussion into our final version. Besides, we are particularly grateful to the reviewer for the suggestion on tackling Q3. In the following, we address the reviewer's further comments on Q1 and Q3:
> > >
> > > **1. Further Response to Q1**
> > >
> > > **Response:** We thank the reviewer for the clarification of the convergence rate and the learning rate choice for NTK. We make the following remarks regarding the comparison of our analysis with standard NTK analysis:
> > >
> > > * First, since our analysis does not have the same model settings and applicability as the traditional NTK analysis, a parallel comparison is not possible. For example, the standard NTK analysis cannot be easily applied to our mildly-parameterized case in which the network width $m=\Omega(\log(n/\delta))$, while our analysis cannot recover all results of NTK due to the assumptions on the data. Hence, the comparison is rather philosophical, showing that our results do not deviate much from standard results. A slower rate does not mean our theory is worse than the traditional NTK theories, while a faster rate also does not mean better.
> > >
> > > * The reviewer mentioned that comparing our results with the worst case of NTK theory is unfair. However, it is fair to assume that an average-case target function dealt with by the NTK analysis has a component along the bottom eigenspace. As long as this happens, the lowest eigenvalue controls the speed of convergence.
> > >
> > > * Our analysis has stronger assumptions on the target function. But it is unfair to compare our results with NTK results only on these target functions, because the assumptions on the model width are very different. For highly over-parameterized models, traditional NTK analysis may indeed produce comparable or even better bounds than those given by our analysis. This does not mean traditional NTK theories beat our theory--our analysis works for under-parameterized model, in which traditional NTK analysis fails. Again, our analysis is a complimentary of NTK theories which can deepen our understanding to the NTK.

---

> > > > ### Comment · Reviewer_uRDi · 2022-08-08
> > > > **Response part 1**
> > > >
> > > > As for the response to Q1, I have some small further disagreements here however I don't want to beleaguer the point to much.  I don't want the area chair to place too much emphasis on this disagreement in the final evaluation.  Even if I *personally* believe NTK tools could prove $\Omega(1)$ descent of the loss at the beginning of training for mildly parameterized networks that doesn't mean the present work is not valuable for providing a novel analysis.  Also even though this may be possible I agree with the authors that the literature does not address that setting to my knowledge and thus it would not be fair to dismiss the present work based on a proof that hasn't been written yet.
> > > >
> > > > Nevertheless, I will explain further what I am thinking of.  Using Bernstein's inequality one can show that deviations of the shallow NTK at initialization are $O(\log(n^2 / \delta) / \sqrt{m})$ (similarly to how you show convergence for the activation patterns).  The typical NTK regime analysis will require that deviations of the NTK are smaller than $\lambda_{min}(H^\infty) / n$.  Thus in the typical setting indeed you need at least $m \gtrsim n^2$.  However one can get away with weaker requirements if one only wishes to analyze the convergence along higher eigenspaces.  For analyzing the top $k$ eigenspaces you only need that deviations of the NTK are small relative to $\lambda_k(H^\infty) / n$ which for fixed $k$ is $\Omega(1)$.  For smooth activations (result notably excludes ReLU) deviations of the NTK within a ball of radius $R$ around initialization are $O(poly(R) / \sqrt{m})$ (see [1]).  Up to an $O(1)$ stopping time, the parameters stay within an $O(1)$ ball around the initialization and thus you can set $R = O(1)$.  Thus if one assumes that the target has $\Omega(1)$ projection onto the top $k$ components then you can demonstrate that the loss will descend along those components.  For networks with bias or networks with asymmetric activation functions like ReLU, the top eigenspace of the NTK significantly aligns with a constant vector.  Thus any target function with nonzero mean will have significant projection onto the top spaces (as you mention in  a comment to one of the reviewers for one hot labels since each label is positive you have significant convergence since it's non-centered).  Even if the target has nonzero projection onto the bottom eigenspace, this is not a problem as long as it has a projection in the top eigenspaces.  Also note that while typical NTK analysis requires overparameterization, NTK analysis up to a stopping time can be done without overparameterization (in fact it can be done in the underparameterized case).
> > > >
> > > > I will emphasize that this exact sort of analysis has not been written down in the literature and the devil is in the details and there may be some details that prevent or complicate such a proof, however to clarify for the benefit of the authors and other readers I found it worth to outline my thoughts.  Again this is more of a personal quibble and I do not want the area chair to place much emphasis on this.
> > > >
> > > > [1] Liu et al. On the linearity of large non-linear models: when and
> > > > why the tangent kernel is constant arxiv: https://arxiv.org/pdf/2010.01092.pdf

---

> > > > > ### Author Response · Authors · 2022-08-09
> > > > > **Thank you for being affirmative**
> > > > >
> > > > > We thank the reviewer for being affirmative on the theoretical contribution of our work. The proof sketch provided by the reviewer does seem very promising to show a similar results as ours. But since it has not been written down, our work can serve as a first attempt to provide this type of results. Adding details to the reviewer's sketch is an interesting future work to do. If the proof does work, then new insights may be obtained by connecting our proof with the more standard NTK analysis.

---

> > ### Comment · Reviewer_uRDi · 2022-08-05
> > **Response to initial author comment (part 2)**
> >
> > In total, my main concern is your response to Q3 about performing the proof as if the optimization problem is smooth (which it is not for ReLU).  To a lesser extent I am a little bit concerned about your comparisons to NTK analysis in response to Q1.  I do think there is value to your unique proof that differs from standard NTK deviation analysis.  However, it is important to note that you are still using the NTK to lower bound the loss gradient.  Also since you are assuming a strong structure for the target function it is not a fair comparison to compare to worst-case estimates that do not assume anything about the target function.
> >
> > [1] Arora, Sanjeev, et al. Fine-grained analysis of optimization and generalization for overparameterized two-layer neural networks. International Conference on Machine Learning, 2019.
> >
> > [2] Gradient Descent Provably Optimizes Over-parameterized Neural Networks by Du et al.
> >
> > [3] Ji, Ziwei and Telgarsky, Matus. Polylogarithmic width suffices for gradient descent to achieve arbitrarily small test error with shallow relu networks. arXiv preprint arXiv:1909.12292, 2019
> >
> > [4] Lee et al. Wide Neural Networks of Any Depth Evolve as Linear Models Under Gradient Descent arxiv: https://arxiv.org/pdf/1902.06720.pdf

---

### Official Review · Reviewer_nRXr · 2022-07-10

**Rating:** 6
**Confidence:** 3
**Soundness:** 3 good
**Presentation:** 3 good
**Contribution:** 3 good

**Summary:**

This paper investigates the dynamics of ReLU neural networks in the initial portion of training. By using the discontinuity of the ReLU activation function on the hidden layer as well as a structured dataset a partitioning of the hidden layer neurons is defined for each datapoint. Specifically hidden neurons are grouped into one of four partitions based on whether their input-to-hidden (hidden-to-output) weights have positive/negative dot product with the input (output). From this structured initial setup it is possible to partition the initial hidden neurons based on the initial weight distributions. This then allows for bounds on the network error to be defined since the dynamics of the partitioned subnetworks are more deterministic and have discernible properties (such as most neurons remaining in their original partition). Thus, the primary results of this work are theoretical statements on the lower bound which the error will decrease by in the early stages of learning and a global convergence theorem for the structured dataset.

**Questions:**

## Main Text
I have the following questions or minor points which could be clarified (in addition to the questions which fit in with the Strengths/Weaknesses above):
* Line 55: two full-stops after "neuron"
* Line 58: It is mentioned that the loss landscape will be complicated at initialization, however, with small initial weights this does not appear to be true as the model will be sitting very close to a smooth saddle point at 0.
* Line 95: It says capital letters are used to denote vectors but the example shows a lower case bold letter for a vector (which is used in the rest of the work)
* Line 131 to 134: As a theoretical tool using the labels to preprocess data is fine (and I acknowledge that this assumption is not new to this work), but it limits the scope of the theory as it then cannot apply to unlabelled data or test data. Again I think this could be mentioned.
* Assumption 4.5: it never becomes apparent to me why this assumption is necessary or different from Assumption 4.1. I would just like to understand that better
* Line 169: Can remove "the"
* Line 216: I assume the $\lambda$'s are the singular values but we should be told what they are.
* For Theorem 5.3: Are only the input-to-hidden weights being trained?

## Appendix (optional)
* Line 519: $0 \leq s \leq t+1$ is stated twice
* Line 520: Where does the $\gamma_1$ variable come from or how is it determined?
* Line 539: what is $P^\infty$, if it is the infinite width limit for the hidden layer then does this fit in with the notion of this theory being for mildly over-parametrized models?
* Line 541: How does the following line apply from the union bound? Is it the union bound over the hidden neuron distributions?
* Line 547: Is this mean to say "union bound"?
* Line 560: Is the final piece of this line guaranteed since from the line before the $-1-f(x,\theta)$ make it seem like the entire second part of the expression could be negative?

As a guide, if points 2 and 3 from the points under weaknesses for Quality are addressed I would be inclined to increase my score by 1. If point 1 is addressed and the results are compelling I would increase my score by 1.

**Limitations:**

There appear to be a couple limitations which could be acknowledged that I have pointed out above. If I am not mistaken then I think they should be mentioned.

**Strengths And Weaknesses:**

# Strengths
## Originality
To my knowledge this is the first work which has theoretically used neuron partitioning, however, as the paper acknowledges, this has previously been done empirically. Focusing on the early stages of learning as opposed to the more prominent approach of investigating the late stages of learning also adds to the originality.

## Quality
The paper is of a high quality and well structures. The research itself is focused in its goal and the methodology appears appropriate. The only figure in the paper is clear and helpful.

## Clarity
The paper is clear and well written. Notation is intuitive, used sparingly and used consistently which aids in the readability. Figure 1 adds to the clarity of the work and displays a helpful setting that demonstrates how Lemma 6.2 can be consistent for different datapoints.

## Significance
This appears to be the most difficult part to assess and I do have some reservations (see below). However, a tighter bound on the gradient norm is given and the theoretical technique of partitioning the hidden neurons could potentially yield future insight. For me the rigor of this work is what adds to the significance substantially. Having gone through Appendix A quite carefully and Appendix C (it seems the proof technique is similar for all Appendices) the proofs seem correct and rigorous.

# Weaknesses
## Originaity
This work does bear some resemblance to prior theoretical work on gated neural networks, see [1] for example. I think this should be acknowledged.

## Quality
I have 3 primary concern from the main text of this work
1. This work could benefit greatly from a simulation of the derived equations. While I think purely theoretical work is certainly acceptable, the theory should still be checked against even a toy example. The XOR dataset [2] for example seems like it would be appropriate and provide a sense of how tight the derived bounds really are. Adding something like this is my main recommendation and its absence my main reservation.
2. On line 98 and 99 the "less than or equal to some multiple" operator $\lesssim$ is defined. This operator appears very loose since $a \lesssim b$ translates to $a \leq cb$ for some positive c value. This appears to be the same as saying $a \leq \infty$ assuming b is positive. As far as I can see it is used sparingly. However, it is used early on in the main text to define the learning rate which is obviously an important variable and impacts other variables such as $\kappa$. It is also used for Lemma 6.4 which is used for the global convergence Theorem. What is the effect of this on the tightness of the bounds and perhaps it is necessary to address this.
3. The learning rate is another point of concern in that some learning rates presented should be unstable, yet no mention is made of this in the text. Specifically if the learning rate must be less than a particular ratio of the top eigenvalue of the hessian to avoid divergence [3]. Such a limitation is not place when defining the learning rate (as mentioned for the second point above) or for Theorem 5.2 where the learning rate is inversely proportional to the loss (and will tend to infinty as a result). I don't think ignoring stability in this case is an issue, however, I again think it should be mentioned.

## Significance
The three points under quality above also limit the  significance, or rather limit my ability to determine this work's significance. The main issue again being that it is unclear how tight the bounds really are or how applicable they are without some form of simulation being run. If not such simulation fits all of the assumptions then this would need to be acknowledged. If this is the case I do not view it as grounds for rejection but it is important for a reader to know. That said, this work seems to be a good addition to the theoretical literature and I think the strength outweigh the weaknesses for its significance.

[1] Lakshminarayanan, Chandrashekar, and Amit Vikram Singh. "Deep Gated Networks: A framework to understand training and generalisation in deep learning." arXiv preprint arXiv:2002.03996 (2020).

[2] Brutzkus, Alon, and Amir Globerson. "Why do larger models generalize better? A theoretical perspective via the XOR problem." International Conference on Machine Learning. PMLR, 2019.

[3] Cohen, Jeremy M., et al. "Gradient descent on neural networks typically occurs at the edge of stability." arXiv preprint arXiv:2103.00065 (2021).

---

> ### Author Response · Authors · 2022-08-02
> **Response to Reviewer nRXr (Part IV)**
>
> **Q6. Line 95: It says capital letters are used to denote vectors but the example shows a lower case bold letter for a vector (which is used in the rest of the work)**
>
> **Response:** We apologize for the confusion. We want to amend this sentence to: the bold letters represent vectors or matrices, such as $\boldsymbol{x}$ and $\mathbf{P}$. And we will modify this in our revised version.
>
> **Q7. Line 131 to 134: As a theoretical tool using the labels to preprocess data is fine (and I acknowledge that this assumption is not new to this work), but it limits the scope of the theory as it then cannot apply to unlabelled data or test data. Again I think this could be mentioned.**
>
> **Response:** We acknowledge that this assumption is important to our theory. However, a complete analysis under such a relatively strong assumption is also of great interest to understand the neural networks' training dynamics. We will work to relax this assumption conditionally, such as linearly separable data. However, our preliminary experiments suggest that the training accuracy may fall into some ``plateau'' during training and that the training dynamics will be more complex. As for a more in-depth study, we leave it to future work.
>
> **Q8. Assumption 4.5: it never becomes apparent to me why this assumption is necessary or different from Assumption 4.1. I would just like to understand that better.**
>
> **Response:** In Section 4.1, the label is $1$ or $-1$, and learning these data representations may be difficult for the neural network (initialized close to 0). So we need stronger data Assumption 4.1 to analyze neural networks' learning process. However, in Section 4.2, our labels are one-hot encoded; thus, each label component is non-negative. In this case, neural networks (initialized close to 0) will quickly learn this bias toward positive, and the loss will drop significantly in the early stage. So we only need to make milder Data Assumption 4.5, and even most of random data can satisfy this assumption. It also verifies the common phenomenon of early stage convergence in practice, especially with one-hot encoding.
>
> **Q9. Line 216: I assume the $\lambda$'s are the singular values but we should be told what they are.**
>
> **Response:** $\lambda$'s are indeed the singular values. In the revised version of the paper, we will add a definition of $\lambda$ in section 3.1.
>
> **Q10. For Theorem 5.3: Are only the input-to-hidden weights being trained?**
>
> **Response:** Yes, compared to training all layers in Theorem 5.2, we consider training only the input-to-hidden weights $\boldsymbol{b}_k$ $({k\in[m]})$ to obtain a faster convergence rate in Theorem 5.3. And it is a common simplification when analyzing two-layer neural networks [6].
>
> **Q11. Line 539: what is $\text{P}^{\infty}$, if it is the infinite width limit for the hidden layer then does this fit in with the notion of this theory being for mildly over-parametrized models?**
>
> **Response:** $\text{P}^{\infty}$ is indeed the infinite width limit for the hidden layer, but this is only an auxiliary variable that does not contradict our theory. Specifically, we want to estimate $card\big(\mathcal{TL}_i(0)\cap\mathcal{TL}_j(0)\big)$ at the initial time. With the help of this limit matrix, we can show that for \textbf{mildly parameterized} neural networks with $m=\Omega(\log(n/\delta))$, with high probability $1-\delta$, $card\big(\mathcal{TL}_i(0)\cap\mathcal{TL}_j(0)\big)>0$  (please refer to Lemma A.2, Line 533). Although some other theories, such as NTK [6], also introduce this matrix, they are not the same as what we are trying to describe. In NTK regime, they would show that neural networks keep close to the infinitely wide network during the training process, so they often need to be \textbf{highly over-parameterized} to get there. However, our theory only needs the neural networks are mildly parameterized, and neural networks' performance throughout the training process may be very different from that of the infinitely wide network, which is the main difference between our theory and NTK.
>
>
> **Q12. For grammar mistakes and typos.**
>
> **Response:** We apologize for the confusion caused by the  grammar mistakes and typos. We will carefully read through the whole paper and correct all typos.
>
>
> &nbsp;
> &nbsp;
> ----
>
> **Reference**
>
> [6] Arora, Sanjeev,et al . Fine-grained analysis of optimization and generalization for overparameterized two-layer neural networks. International Conference on Machine Learning, 2019.

---

> > ### Comment · Reviewer_nRXr · 2022-08-07
> > **Rebuttal Response  (Part IV)**
> >
> > Q7. I just want this to be mentioned in the main text as a limitation. The assumption itself is reasonable.
> >
> > Q8. I think my points on MNIST and similar datasets above apply here too, so I will not reiterate. But even your response to the question above would be a valuable addition to the paper. Again, what I would look for to increase my score is for you to add context and justification to the steps in the work as well as being more clear on the limitations.
> >
> > Q10. This needs to be made more clear. In Theorem 5.3 it even says "let $\{b_k(t)\}_{k\in[m]}$ be the parameters of model (4)". To recognize that these are only the input layer weights requires a lot of attention and for the reader to look back to the top of page 4 to check the model definition, just to realize that the theorem is not completely precise in its statement. It seems easy to say "...be the input layer parameters of model (4)". I appreciate that this is a theoretical work and great care was taken with the notation (my original review pointed this out as a positive) but relying too heavily on (even good) notation makes a paper difficult to follow or to understand confidently. I hope I have given enough examples and explained myself clearly enough so that the authors are aware of my broader point and are able to incorporate it into their revised version. I do believe it will make for a stronger paper that is of interest to the broader ML community.
> >
> > Q11. Mentioning what $P^\infty$ is would be helpful. Thank you for the explanation on the difference between the paradigm of this work and the NTK. I would suggest rethinking the terminology of "mildly parametrized" models. Perhaps I am missing something and this is a known terminology, but myself and another reviewer called it "mildly over-parametrized" and your citations also seem to use that terminology. While I understand the distinction to be that mildly parametrized models is a broader set of models (being mildly over-parametrized models **and** under-parametrized models) the terminology of "mildly parametrized" does not seem intuitive for this set of models.

---

> ### Author Response · Authors · 2022-08-02
> **Response to Reviewer nRXr (Part III)**
>
> **Q3. The learning rate is another point of concern in that some learning rates presented should be unstable, yet no mention is made of this in the text. Specifically if the learning rate must be less than a particular ratio of the top eigenvalue of the hessian to avoid divergence [2]. Such a limitation is not place when defining the learning rate (as mentioned for the second point above) or for Theorem 5.2 where the learning rate is inversely proportional to the loss (and will tend to infinty as a result). I don't think ignoring stability in this case is an issue, however, I again think it should be mentioned.**
>
> **Response:**
> - The reviewer is correct that the learning rate should be upper bounded by the quantity related with the top eigenvalue of the Hessian. Our analysis also considered this factor, but implicitly.
>
> - For global convergence results (Theorem 5.2 and 5.3), the loss function we consider is **exponential-type** (Assumption 5.1, Line 209), such as exponential loss $\ell(z)=e^{-z}$ and logistic loss $\ell(z)=\log(1+e^{-z})$, which has a nice property that $||\nabla^2 \mathcal{L}(\boldsymbol{\theta})||_2$ can be controlled by the loss $\mathcal{L}(\boldsymbol{\theta})$ along GD trajectory. Specifically, for Theorem 5.2, there exists an absolute constant $C>0$, s.t. $||\nabla^2 \mathcal{L}(\boldsymbol{\theta}(t))||_2\leq Ct\mathcal{L}(\boldsymbol{\theta}(t))$ for any $t\geq0$, please refer to Lemma C.15 (Line 985); for Theorem 5.3, there exists an absolute constant $C'>0$, s.t. $||\nabla^2 \mathcal{L}(\boldsymbol{\theta}(t))||_2\leq C'\mathcal{L}(\boldsymbol{\theta}(t))$ for any $t\geq0$, please refer to Lemma D.12 (Line 1088). The similar property of exponential-type loss has also been discovered and proved in related theoretical papers: Lemma 6 in [3] (P29) and Lemma E.7 in [4] (P32). Therefore, we can transform the Hessian-controlled learning rate condition into the loss-controlled learning rate condition. So we obtain the learning rate condition $\eta_t=c/t\mathcal{L}(\boldsymbol{\theta}(t))$ in Theorem 5.2 and $\eta_t=c/\mathcal{L}(\boldsymbol{\theta}(t))$ in Theorem 5.3 $(c\leq 1/2)$, which can guarantee GD convergence.
>
> - As for the early stage convergence results (Theorem 4.2 and 4.6), since the parameters do not go very big at the early stage, we can obtain uniform estimate of Hessian over all possible parameters. Please refer to Lemma B.13 (Line 792) for details. Therefore, we can use the small constant learning rate $\eta\leq 0.01$ to ensure loss descent in this stage.
>
>
> **Q4. This work does bear some resemblance to prior theoretical work on gated neural networks, see [5] for example. I think this should be acknowledged.**
>
> **Response:** We thank the reviewer for pointing this interesting related work [5]. For understanding ReLU activation, the neural partition approach in our article (true/false, living/dead) is very similar to the view of gating networks (0-1) in [5]. And we believe this work is very enlightening for extending our two-layer theory to deep ReLU neural networks. We will add and discuss this work in the revised version.
>
>
> **Q5. Line 58: It is mentioned that the loss landscape will be complicated at initialization, however, with small initial weights this does not appear to be true as the model will be sitting very close to a smooth saddle point at 0.**
>
> **Response:** First, we agree with the reviewer that using a small initialization does make the network close to the saddle point at 0, and the loss landscape is clear at initialization. However, analyzing how GD (without any noise) escapes the saddle point region at initialization is still a complex and meaningful problem. Our analysis points out the following training dynamics through our neuron partition technique. Under some data assumption, with a proper learning rate $\eta$ and small initialization scale $\kappa$, GD (without any noise) will escape from this saddle region in the first step, then enter a good region with neither spurious local minima nor saddle points. In the revised version, we will fix this point and add a more accurate explanation.
>
>
> &nbsp;
> &nbsp;
> ----
>
> **Reference:**
>
> [2] Cohen, Jeremy M., et al. Gradient descent on neural networks typically occurs at the edge of stability. arXiv preprint arXiv:2103.00065, 2021.
>
> [3] Chatterji, Niladri S, et al. When does gradient descent with logistic loss find interpolating two-layer networks? J. Mach. Learn. Res., 2021.
>
> [4] Lyu, Kaifeng, et al. Gradient descent maximizes the margin of homogeneous neural networks. arXiv preprint arXiv:1906.05890, 2019.
>
> [5] Lakshminarayanan, Chandrashekar, et al. Deep Gated Networks: A framework to understand training and generalisation in deep learning. arXiv preprint arXiv:2002.03996, 2020

---

> > ### Comment · Reviewer_nRXr · 2022-08-07
> > **Rebuttal Response (Part III)**
> >
> > Q3. Thank you for the clarification on this point. I think it may be helpful mentioning that stability is implied by the given setting and equations.
> >
> > Q4. Thank you. I look forward to the revised edition.
> >
> > Q5. This was a small point on the statement being made. I appreciate that the dynamics out of the saddle at $0$ is a meaningful problem.

---

> ### Author Response · Authors · 2022-08-02
> **Response to Reviewer nRXr (Part II)**
>
> **Q2. On line 98 and 99 the ``less than or equal to some multiple'' operator $\lesssim$ is defined. This operator appears very loose since $a\lesssim b$ translates to $a\leq c b$ for some positive $c$ value. This appears to be the same as saying $a\leq\infty$ assuming $b$ is positive. As far as I can see it is used sparingly. However, it is used early on in the main text to define the learning rate which is obviously an important variable and impacts other variables such as $\kappa$. It is also used for Lemma 6.4 which is used for the global convergence Theorem. What is the effect of this on the tightness of the bounds and perhaps it is necessary to address this.**
>
> **Response:**
> - The notation $a\lesssim b$ is much stronger than saying $a$ is finite. For $a,b>0$, we use the notation $a\lesssim b$ to indicate that there exists an **absolute** constant $c > 0$ such that $a\leq cb$, which means that (i) if $b$ has a upper bound, $a$ will be bounded; (ii) if $b$ goes to $0$, $a$ will also go to $0$. So it is different from saying $a\leq\infty$ assuming $b$ is positive.
>
> - Actually, we are able to conduct more explicit estimates of important parameters such as the learning rate $\eta$. (i) For the early stage convergence results (Theorem 4.2 and 4.6), our proof holds not only for $\eta=0.01$ but also for any $\eta\leq 0.01$ due to the stability of Gradient Descent, as mentioned by the reviewer. Actually, we have proved one of the upper bounds of $\eta$ in Theorem 4.2 and 4.6 is 0.01. (ii) For the global convergence results (Theorem 5.2 and 5.3), we have precise estimate of the learning rate $\eta_0$ in the Appendix, i.e. $\eta_0\leq 1/2\sqrt{2}$ in Theorem C.17 (Line 1002) and Theorem D.13 (Line 1097). We will eliminate these $\lesssim$ related to $\eta$ in the new version.
>
> - Due to limited space, we choose to use the asymptotic notation instead of precise estimates. But many of the relations can be estimated accurately. For example, (i) the constants hidden by $\lesssim$ in the definition of $T_e$ (Line 274) are $1000000/251001(1/2+2\sqrt{\log(2n^2/\delta)/m})$ and $2\sqrt{2}$, shown in Line 581; (ii) the constant hidden by $\gtrsim$ in Lemma 6.4 (Line 291) is $\frac{999}{1000}$, proved in Lemma A.10 (Line 630); (iii) the constant hidden by $\gtrsim$ in Lemma 6.7 (Line 318) is proved in Lemma C.10 (Line 942). In the revised version of the paper, we will try to replace $\lesssim$ with their precise estimates at least in the Appendix.

---

> > ### Comment · Reviewer_nRXr · 2022-08-07
> > **Rebuttal Response (Part II)**
> >
> > I am aware that there is a difference, my critique is in the context of how the notation is used in this work. My apologies if that was unclear. The definition of $T_e$ (Line 274) is a good example of my concern. Here the notation is used to denote a term as being "less than or equal to some multiple" of 1. Clearly $b$ going to $0$ does not apply here. What remains is the difficulty of telling from this theorem what a reasonable $c$ value would be. In this case it seems to be a reasonably sized value (depending on hyper-parameters) but there is no way to tell from the main text.
> >
> > I am not certain that doing away with the notation is helpful. I agree that the space constraint limits adding in the actual values. The actual values themselves are also complex and may not help a reader contextualize what is being said anyway. I think the best way to deal with this is to mention what sort of scale the hidden constant may be. As you point out you already prove or show many of these values in the Appendix but for a reader going through the main text it is important that they understand what these equations are saying. I appreciate that Corollary 4.3 and 4.7 aim to serve this role but I feel that there are still small comments which can be made to make the results or setup clearer. For example mentioning when MNIST applies (or doesn't like it in global convergence case) or mentioning the scale of the hidden constants when relevant.

---

> > > ### Author Response · Authors · 2022-08-09
> > > **Response to Q2**
> > >
> > > We thank the reviewer for providing an excellent way to deal with the use of $\lesssim$. Indeed, it is helpful to give the scale of the hidden constant if the constant is complex. In our revised version, we fixed this issue by either providing the exact hidden constant or giving its scale, such as Line 146, Line 298, and Line 346 in the revised paper.

---

> ### Author Response · Authors · 2022-08-02
> **Response to Reviewer nRXr (Part I)**
>
> **Q1. This work could benefit greatly from a simulation of the derived equations. While I think purely theoretical work is certainly acceptable, the theory should still be checked against even a toy example. The XOR dataset [1] for example seems like it would be appropriate and provide a sense of how tight the derived bounds really are. Adding something like this is my main recommendation and its absence my main reservation.**
>
> **Response:**
> We thank the reviewer's suggestions on experiments and dataset. We did two experiments to show the effectiveness of our theoretical results. The experimental results are shown and discussed in the following.
>
> - **(1). Early Stage Convergence (Theorem 4.6).** Our theory about early stage convergence applies to a wide range of dataset (please refer to Assumption 4.5 in Section 4.2). We use the following setup to compare the experimental result with our Theorem 4.6.
>     - Dataset: the **first 1000** data in **MNIST** dataset with normlization $\left\|\|\boldsymbol{x}\right\|\|_2=1$. (It satisfies Assumption 4.5.)
>     - Loss: the logistic loss $\ell(\boldsymbol{y}_1,\boldsymbol{y}_2)=\log(1+\exp(-\boldsymbol{y}_1^\top\boldsymbol{y}_2))$. Network: the two-layer ReLU network. $m,\kappa,\eta$ are choosen by Theorem 4.6.
>
>     We study the change of our bounds under different network width $m$ and learning rate $\eta$. In the first experiment, we fix $\eta=0.01$ and change $m$; in the second experiment, we fix $m=200$ and change $\eta$.
>
> |  |  |  |  |  |
> | :-:| :-: | :-: | :-: | :-: |
> | **$m$** | 100 | 200 | 500 | 1000 |
> | __Our hitting time__ | 34 | 34 | 34 | 34 |
> | __Realistic loss descent__ |$1.96\times10^{-1}$| $2.68\times10^{-1}$ | $4.74\times10^{-1}$ | $5.71\times10^{-1}$ |
> | **Our loss descent bound** |$0.65\times10^{-1}$|$0.69\times10^{-1}$| $0.74\times10^{-1}$ | $0.81\times10^{-1}$ |
>
> |  |  |  |  |  |
> | :-:| :-: | :-: | :-: | :-: |
> | $\eta$ | 0.01 | 0.005 | 0.002 | 0.001 |
> | __Our hitting time__ | 34 | 69 | 173 | 346 |
> | __Realistic loss descent__ |$2.68\times10^{-1}$| $2.91\times10^{-1}$ | $2.86\times10^{-1}$ | $2.88\times10^{-1}$ |
> | __Our loss descent bound__ |$0.69\times10^{-1}$|$0.69\times10^{-1}$| $0.69\times10^{-1}$ | $0.69\times10^{-1}$ |
>
>     The results above shows that our loss descent bound (4-th row in each table) is relatively tight, basically in the same order of magnitude as the realistic loss descent (3-rd row in each table).
>
> - **(2). Global Convergence (Theorem 5.3).** Since the XOR dataset [1] does not satisfy our Assumption 4.1, we conduct this part of experiments on a synthetic dataset that we construct.
>
>     - Dataset: We use the following distribution to generate data: For the class $y=1$, $\boldsymbol{x}=(\cos\theta,\sin\theta,1_{d-2}^{\top})^{\top}/\sqrt{d-1}\in\mathbb{R}^d$, where $\theta\sim\mathbb{U}([0,2\pi])$; For the class $y=-1$, $\boldsymbol{x}=(\cos\theta',\sin\theta',-1_{d-2}^\top)^\top/\sqrt{d-1}\in\mathbb{R}^d$, where $\theta'\sim\mathbb{U}([0,2\pi])$. It is easy to verify this distribution satisfies our Assumption 4.1 if $d\geq 3$.
>
>     In this experiment, we let $d=50$, then generate $1000$ data with $y=1$ and $1000$ data with $y=-1$. We use the increasing learning rate in Theorem 5.3, and we will compare our convergence rate in Theorem 5.3 with the realistic convergence rate. Due to limited space, we can show the comparison at a few iterations in the following table. In the revised version, we will show the complete comparison through figures.
>
>
> |  |  |  |  |  | |
> | :-:| :-: | :-: | :-: | :-: |  :-: |
> | **Iteration** | 0 | 1000 | 2000 | 3000 | 4000 |
> | **Realistic loss** | $6.93\times10^{-1}$ | $2.04\times10^{-3}$ | $2.93\times10^{-5}$ | $8.65\times10^{-7}$ | $1.26\times10^{-8}$ |
> | **Our loss bound** |$6.93\times10^{-1}$|$9.81\times10^{-2}$| $1.39\times10^{-2}$ | $1.97\times10^{-3}$ | $2.78\times10^{-4}$ |
>
>     From the results above, our loss bounds are not very tight in this case. But at least the convergence of GD is linear, which corresponds to our theoretical results. We acknowledge that more refined analysis may potentially improve the tightness of the bounds. This may be a direction of future work.
>
> &nbsp;
> &nbsp;
> ----
>
> **Reference**
>
> [1] Brutzkus, Alon, et al. Why do larger models generalize better? A theoretical perspective via the XOR problem. International Conference on Machine Learning, 2019.

---

> > ### Comment · Reviewer_nRXr · 2022-08-07
> > **Rebuttal Response**
> >
> > Thank you to the authors for running these experiments. I think they are very insightful for understanding the theoretical results of this work. I would like to emphasize that my concerns with this work are not theoretical (having read your discussion with Reviewer uRDi). I think this work is valuable and interesting. Similarly the fact that the above experiment on the global convergence showed that the bound is not exceptionally tight does not bother me. What I find troubling is that I would never have known this without the simulation. Thus, if I were to  build on this work I would not know where the open directions were and it would not have been clear how to proceed (without first running my own simulations). I see my fellow reviewers have also commented on the fact that the limitations are not clear, but I think this is due to a broader problem. Namely, that it is extremely difficult to _contextualize_  this work. In hind-sight it is obvious that MNIST applies in Section 4.2, but while reading the work initially I was not thinking about MNIST (or a similar dataset). Again, running a simulation on a recognizable dataset helps with this.
> >
> > I do have a follow-up question. Assumption 4.5 holds for any normalized dataset with no explicit contradictions (completely oppose input directions mapping to same label). This seems true of almost all datasets, or even just image datasets. So would the theory hold for something as complex as CIFAR-10 or ImageNet?
> >
> > I look forward to the revised version of the paper that includes the added simulations and analysis.

---

> > > ### Author Response · Authors · 2022-08-09
> > > **Response to Q1**
> > >
> > > We thank the reviewer for recognizing the value of our experiments and providing a valuable follow-up question. We believe that  Assumption 4.5 indeed holds for almost any image dataset, such as CIFAR-10 and ImageNet. In our revised version, we add an explanation regarding this issue under Assumption 4.5. We also added a new experiment using the CIFAR-10 data to verify the bounds in Theorem 4.6. The results are given in the Table 1 in **Appendix H** of the revised paper. The early stage convergence phenomenon still exists for CIFAR-10. It worth mentioning that, our loss descent bound actually does not change with the choice of different datasets, so it is pretty robust.

---

> > > ### Author Response · Authors · 2022-08-09
> > > **A revised version is ready**
> > >
> > > We thank again for valuable comments and suggestions by the reviewer. We have uploaded a revised version of the paper, which includes many discussions inspired by the reviewers' comments. Please refer to the file named **"Revision_for_nRXr.pdf"** in "Supplementary Material". In the revised paper, all changes relevant with Reviewer nRXr are marked **orange**. Changes according to other reviewers are marked red.
> > >
> > > Because of the 9-page limit for revisions during the rebuttal/discussion period, we have to put some important contents into the Appendix, such as the experimental results (Appendix H) and Figure 1 (Appendix A). We will try to improve the structure of the paper and include those important contents in the main paper in the camera-ready version, if we have the chance.

---

### Official Review · Reviewer_rRrN · 2022-07-13

**Rating:** 5
**Confidence:** 4
**Soundness:** 3 good
**Presentation:** 2 fair
**Contribution:** 3 good

**Summary:**

This paper studies the training dynamics of GD and SGD in mildly over-parameterized neural networks. Specifically, the major result focuses on the early-stage training dynamics and proves that the loss decreases by a significant amount in the early iterations. Global convergence result is also provided.

**Questions:**

Line 57: Is it "mildly overparameterized" instead of "mildly parameterized"?

**Limitations:**

Yes.

**Strengths And Weaknesses:**

The paper is technically solid. I think the most interesting result is the early-stage result, as it presents a concrete hitting time when the loss descends to a relatively low value. In particular, the hitting time is proved to be small, which matches the empirical observation that the training loss descends rapidly in the first few epochs. Therefore, this early-stage result might be able to explain practical training. The partition technique also looks interesting and may help in understanding the training dynamics of GD in neural nets.

There are still some weak points in this paper.

I skim the appendix and believe the proof is correct. However, the author seems to assume specific values of $\eta$, $\kappa$ and $m$ in the proof. It is better to keep $\eta$, $\kappa$ and $m$ in the complete proof to better observe the relationship between the loss bound and these parameters.

The assumption is still a bit strong. Specifically, the initialization is assumed to be close to 0. This assumption may be not desirable in the following two aspects: 1. It may not characterize the training dynamics in the global region; 2. It may not match practice.

The settings are not unified in binary and multi-class classification. Specifically, for binary classification, the paper uses the ReLU network without bias; whereas for multi-class classification, the paper uses the ReLU network with bias. It would be better if the settings are changed to the same.

The comparison with the results in late-stage training (Line 50, 185) looks a bit strange. In my humble point of view, these results focus on the generalization performance of GD in neural networks and are orthogonal to the topic of this paper.

The writing of this paper could be improved. The partition names in Lemma 6.2 look confusing as “false-dead” neurons are truly dead in the sense that they remain dead, and “true-dead” neurons are not dead as they can turn into true-living. There are also some grammar mistakes in the paper, especially in the singular/plural form and the usage of articles (Line 41, 42, 55, 74, etc.).

---

> ### Author Response · Authors · 2022-08-02
> **Response to Reviewer rRrN (Part II)**
>
> **Q4. The comparison with the results in late-stage training (Line 50, 185) looks a bit strange. In my humble point of view, these results focus on the generalization performance of GD in neural networks and are orthogonal to the topic of this paper.**
>
> **Response:** Our work is twofold as shown by the title. On one hand, we can prove early stage convergence. On the other hand, we can show the global full convergence under some conditions. In these cases, every stage of the training dynamics is characterized, including the later stage. Therefore, it is reasonable to discuss related results on late-stage training. Compared with those works that analyzes the dynamics after the training loss is sufficiently small, our analysis characterizes the whole training process, not only the early and late stages, but also the transition between these stages.
>
> **Q5. The writing of this paper could be improved. The partition names in Lemma 6.2 look confusing as “false-dead” neurons are truly dead in the sense that they remain dead, and “true-dead” neurons are not dead as they can turn into true-living. There are also some grammar mistakes in the paper, especially in the singular/plural form and the usage of articles (Line 41, 42, 55, 74, etc.).**
>
> **Response:** We apologize for the confusion caused by the writing. In the revised version of the paper, we will carefully look over the paper and correct all grammar and spelling problems. Here we want to especially clarify the meaning of "true" in "true-dead". It means the prediction of the $k-$th neuron on data $\boldsymbol{x}_i$ is correct, (i.e. $a_k\sigma(\boldsymbol{b}_k^\top\boldsymbol{x}_i)$ has the same sign as the label $y_i$), and the neuron is "dead", (i.e. $\boldsymbol{b}_k^\top\boldsymbol{x}_i<0$). To reduce confusion, we will use "dead-true" instead of "true-dead" and add more explanations on the terminologies in the revised version.
>
> **Q6. Line 57: Is it "mildly overparameterized" instead of "mildly parameterized"?**
>
> **Response:** Our results hold as long as $m=\Omega(\log(n/\delta))$. In this case, the number of parameters may not exceed the number of data. Hence, it is not necessarily overparameterized, and hence we call it "mildly parameterized". In the revised version of the paper, we will add an explanation of the terminology.

---

> ### Author Response · Authors · 2022-08-02
> **Response to Reviewer rRrN (Part I)**
>
> We thank the reviewer for comments and suggestions to improve this paper. In the following, we answer the reviewer’s questions in detail.
>
> **Q1. I skim the appendix and believe the proof is correct. However, the author seems to assume specific values of $\eta,\ \kappa$ and $m$ in the proof. It is better to keep $\eta,\ \kappa$ and $m$ in the complete proof to better observe the relationship between the loss bound and these parameters.**
>
> **Response:**
> - Our analysis in the proofs of Theorem 5.2 and Theorem 5.3 is general and does not require specific values of $\eta,\ \kappa$, or $m$ (Please refer to Appendix C and D). Foror the proofs of Theorem 4.2 and Theorem 4.6, we use a deterministic $\eta$ for the ease of presenting the results, and the values of $\kappa$ and $m$ are not necessarily deterministic (Please refer to equ(11) (Line 523) and equ(17) (Line 697)).
> - Actually, most part of the proofs of Theorem 4.2 and Theorem 4.6 are for general $\eta$, instead of specific values. Specific values of $\eta$ are only used to estimate an absolute constant (Lemma A.14) at the end of the proof, making the results less hyperparameter-dependent. Moreover, this specific $\eta$ is the taken as the largest $\eta$ that our theory can work. Our analysis works for any $\eta\leq 0.01$ due to the stability of Gradient Descent. In the revised version of the paper, we will restate the theorems to make them more general.
>
> **Q2. The assumption is still a bit strong. Specifically, the initialization is assumed to be close to 0. This assumption may be not desirable in the following two aspects: (1). It may not characterize the training dynamics in the global region; (2). It may not match practice.**
>
> **Response:** The initialization that we considered may not coincide with all practices. But it has been explored extensively in both practical and theoretical studies.
> - In practice, recent studies have shown that the initialization close to 0 achieves good generalization. For example, in [1], the authors propose the ``fixup initialization'', which initializes some convolutional or fully-connected layers at (or close to) 0. Experiments in [1] show that networks with fixup initialization generalize well even without regularization or normalization.
> - Theoretically, close-to-zero initialization is widely used in the analysis of two-layer neural networks [2][3], and such initialization enables a speed separation that helps in the analysis of the training process.
>
> **Q3. The settings are not unified in binary and multi-class classification. Specifically, for binary classification, the paper uses the ReLU network without bias; whereas for multi-class classification, the paper uses the ReLU network with bias. It would be better if the settings are changed to the same.**
>
> **Response:** We thank the reviewer's suggestions on the consistency. In the revised version of the paper, we will unify the settings of the two cases. Actually, our results about the early stage convergence hold for ReLU networks with or without bias. For the binary classification problem, we presented results without bias because in this setting we have a sharper loss descent bound.
>
> &nbsp;
> &nbsp;
> ----
>
> **Reference**
>
> [1] Zhang, Hongyi, et al. *Fixup initialization: Residual learning without normalization.* arXiv preprint arXiv:1901.09321, 2019.
>
> [2] Arora, Sanjeev, et al. *Fine-grained analysis of optimization and generalization for overparameterized two-layer neural networks.* International Conference on Machine Learning, 2019.
>
> [3] Lyu, Kaifeng, et al. *Gradient descent on two-layer nets: Margin maximization and simplicity bias.* Advances in Neural Information Processing Systems, 2021.

---

### Meta-Review · Area_Chair_zF9p · 2022-08-27

**Recommendation:** Accept
**Confidence:** Certain

**Metareview:**

There was an extensive discussion of the article concluding in reviewer recommendations weak accept, borderline accept, and accept. Although there were some reservations, particularly about the assumptions on the data, model and loss, the reviewers found that the article is well-written, technically sound, that it studies important problems of neural network training process, and is valuable for providing a novel analysis. Based on these merits, I am recommending accept. However, I will ask that the authors carefully consider the extensive feedback in the preparation of the final manuscript, particularly the comments concerning the presentation and discussion of the assumptions and limitations, and also carefully work on the improvements discussed during the discussion period, particularly the technical detail of the non Lipschitz gradient, as well as the promised additions, such as the proofs for the quadratic loss in the multi-class setting.


**Award:**

No

---

### Decision · Program_Chairs · 2022-09-14

Accept